# Refining weak supervision for robust lung cavity segmentation: A graph-affinity method with boundary constraints

Zeyu Ding[1], Zhuoyi Tan[2], Hizmawati Madzin[3]*, Zhengdong Li[4], Juntao Liu[2]

**1** Department of Computer Science, Changzhi University, Changzhi, China, **2** School of Low-Altitude Technology and Engineering, Guangdong Polytechnic Normal University, Heyuan, China, **3** Faculty of Computer Science and Information Technology, Universiti Putra Malaysia, Serdang, Malaysia, **4** School of Computer Science and Engineering, Nanyang Technological University, Singapore, Singapore

* hizmawati@upm.edu.my

## Abstract

Pixel-level annotation of lung cavities (LCs) in computed tomography (CT) images is challenging due to their morphological diversity and complexity. Weakly supervised semantic segmentation (WSSS) methods, which utilize sparse annotations (e.g., image-level labels), offer a promising solution. However, existing WSSS approaches often generate coarse pseudo-labels and lack sufficient spatial supervision, resulting in under- or over-segmentation of irregular lesions. To address these limitations, we introduce several key innovations. First, we propose a novel Graph-based Affinity Network (GA-Net) that, unlike conventional methods relying on low-level pixel features, models long-range contextual relationships and structural dependencies using a superpixel graph and learned edge inference kernel, enabling structure-aware pseudo-label refinement for complex lesion morphology. Second, we introduce region-wise affinity propagation, which refines segmentation by propagating activations within semantically coherent 3D regions, offering more precise control over under-/over-segmentation compared to global affinity methods. Additionally, we incorporate Exponential Moving Average (EMA) ensembling for training stability and a scribble-based segmentation module that utilizes pseudo-label contours to provide direct boundary supervision. Extensive experiments on three benchmark datasets demonstrate that our method outperforms existing state-of-the-art medical WSSS techniques, achieving precise and reliable segmentation of complex LCs in CT scans.

## Introduction

Lung cavities (LCs) are crucial radiographic indicators for diagnosing tuberculosis (TB), playing an essential role in confirming diagnoses, assessing disease progression, and monitoring the efficacy of treatment regimens [1,2]. However, due to the

**Data availability statement:** The minimal dataset underlying the conclusions of this manuscript is available in Figshare at https://doi.org/10.6084/m9.figshare.31033282.

**Funding:** Research Management Centre UPM and Faculty of Computer Science and Information Technology, Universiti Putra Malaysia for Journal Publication Grant.

**Competing interests:** The authors have declared that no competing interests exist.

variability in the shape and size of LCs, accurately annotating these LCs at the pixel-level in computed tomography (CT) scan is highly challenging [2]. In recent years, deep learning (DL) has experienced rapid advancements, leading to its widespread adoption in medical image analysis [3–6], particularly in the identification of pulmonary TB LCs [2,7]. In addition, due to the data-hungry nature of DL technology, segmentation models based on fully supervised training paradigms typically require vast amounts of labor-intensive accurate LCs pixel-level annotation data for learning [3–6]. Obtaining these precise pixel-level annotations for LCs often requires a significant amount of human and material resources. Utilizing sparse labels in weakly supervised semantic segmentation (WSSS) methods has become an important trend to overcome this limitation [2,8–10]. These methods train semantic segmentation models using image-level [2,10], scribble [11,12], or bounding box [13] annotations as supervision signals.

Although some weakly supervised segmentation methods perform well in other imaging domains [14–16], they often lack versatility when directly applied to medical lesion tissue segmentation (e.g., TB LCs recognition), making it difficult to achieve good recognition results [2,12,17–19]. In particular, two major challenges often arise when applying existing weakly supervised segmentation methods specifically to TB LCs segmentation research: 1) In the pseudo-label generation stage, many conventional methods [20–24] rely on initial seeds derived from Class Activation Mapping (CAM) [25] and refine them through affinity propagation [21,22,24], such as the random walk [26] or bilateral affinity models adopted in works like IRNet [21]. However, the limitations of these conventional affinity propagation algorithms become pronounced when dealing with LCs, which often exhibit irregular shapes, thin walls, or internal septations. These methods typically construct affinity matrices based on local, low-level pixel appearance similarities (e.g., color and texture), lacking effective modeling of long-range contextual relationships and complex structural dependencies [20–24]. As a result, the propagation process is prone to failure at cavity wall discontinuities or in low-contrast regions, leading to persistent under-segmentation (failure to cover the entire cavity region) or over-segmentation (incorrect inclusion of adjacent normal tissues) in the generated pseudo-labels, as illustrated in Fig 1. 2) At the level of supervision signals, WSSS paradigms that rely exclusively on image-level labels–such as classical CAM-based methods and their variants–provide extremely sparse and weak supervision to the model. Under this setting, models tend to focus on the most discriminative local regions (e.g., the most visually salient parts of a cavity) while failing to perceive and learn the complete spatial extent, precise morphology, and detailed boundary characteristics of the target. Although subsequent studies [27,28] have attempted to alleviate this issue by incorporating self-attention mechanisms or boundary-aware constraints, these approaches may struggle to obtain reliable learning signals when the pseudo-label quality is poor during early training stages, thereby limiting their performance ceiling in complex medical image segmentation tasks.

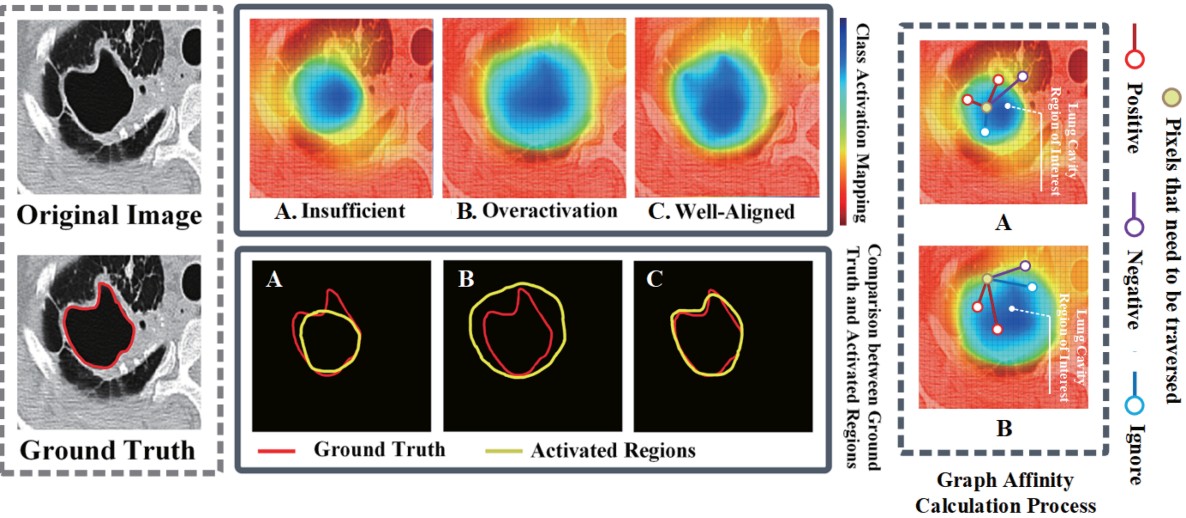

**Fig 1**. **Comparison of recognition results with conventional affinity and graph affinity method.** A: conventional affinity method. B: graph-based affinity methods.

In this paper, we propose a novel WSSS method that introduces targeted innovations to directly address these two core challenges, as shown in Fig 2. Our primary contribution is a new affinity learning paradigm for pseudo-label refinement. Unlike previous methods [20–24], we propose a Graph-based Affinity Network (GA-Net) that constructs a super-pixel graph to represent image structure and employs a trainable network as a kernel to infer affinities, enabling structure-aware, long-range diffusion tailored for complex lesions. Complementing this, we design a Region-wise Affinity Propagation mechanism that refines the CAM by propagating activation scores from high-confidence regions to semantically similar, low-activation regions. This mechanism effectively addresses the problem of coarse pseudo-labels generated by CAM, enhancing the localization of LCs and overcoming the limitations of under-segmentation or over-segmentation. By using a learned affinity matrix, our method leverages regional information to propagate affinities within smaller 3D regions, leading to a more precise segmentation. Additionally, to enhance the robustness of our model, we integrate Exponential Moving Average (EMA) ensembling, which stabilizes model predictions in the early stages of training, reducing the impact of transient errors. This technique smooths the predictions, allowing for better consistency in the refinement of pseudo-labels during training. To further refine the segmentation of small and challenging lesions, we incorporate a scribble-based segmentation module. This module utilizes sparse scribble annotations generated from pseudo-label contours, enhancing the model's ability to capture fine boundary details of small lesions. A partial cross-entropy loss is used to focus the model's attention on the annotated lesion areas, improving its performance in detecting and segmenting small lesions that might otherwise be overlooked. Through extensive experiments on three standard datasets, our proposed method demonstrates superior performance compared to existing state-of-the-art 3D WSSS methods.

## Methodology

**Intuitive overview.** The core objective of GA-Net is to integrate appearance features, spatial context, and the weakly-supervised localization cues provided by the CAM to infer whether two superpixel regions belong to the same anatomical structure. To avoid the high computational cost and noise sensitivity of pixel-wise affinity propagation, the proposed method first partitions the image into superpixels and constructs a compact region-adjacency graph on top of them. In this graph, each node corresponds to a superpixel region, and edges connect region pairs that are either spatially adjacent or feature-similar, thereby encoding potential semantic relationships between them. GA-Net learns the affinity weights of

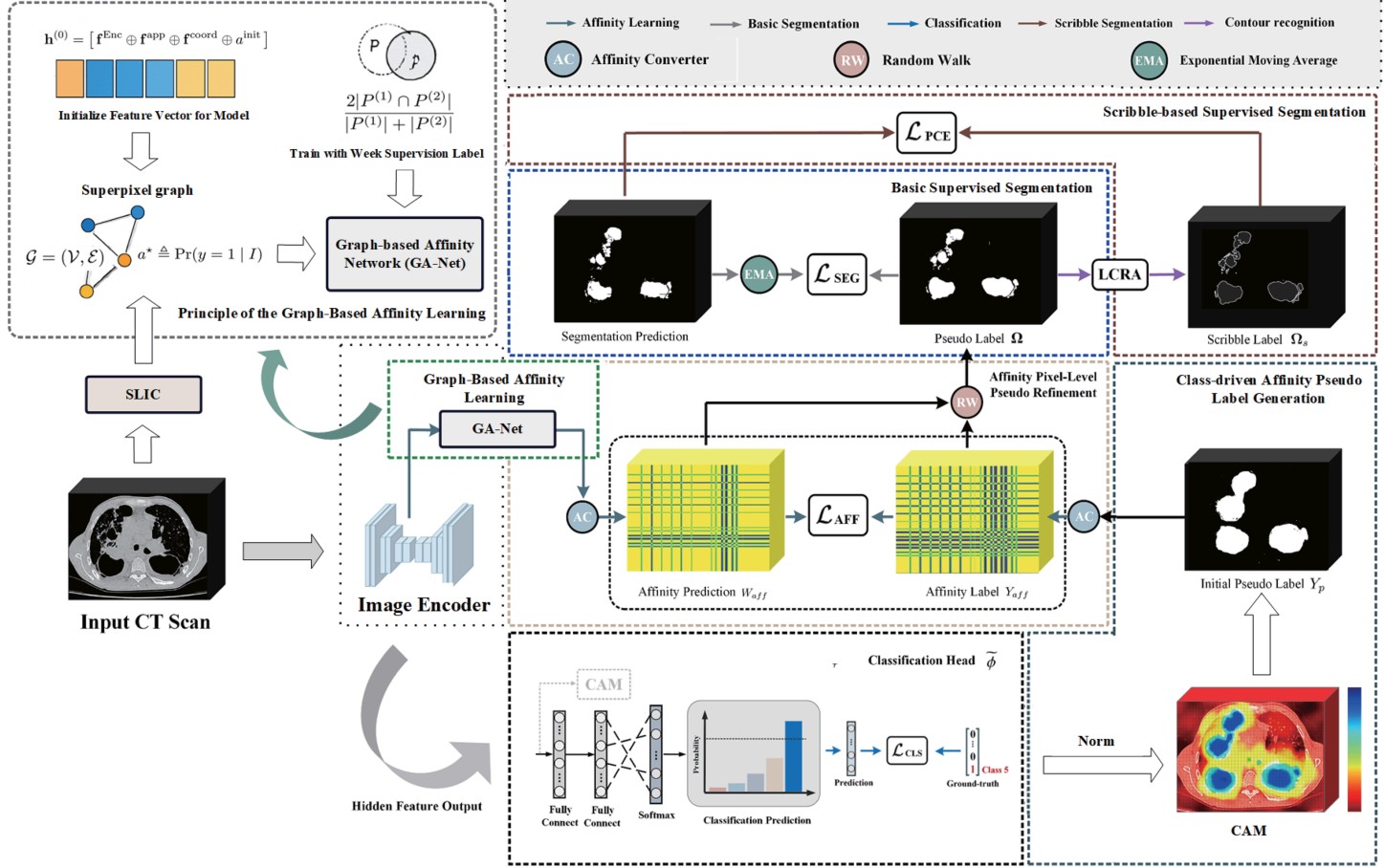

**Fig 2. Overview of the proposed weakly supervised semantic segmentation method.**

the graph edges in an end-to-end, data-driven manner. The supervision signal originates from the soft overlap measure between regions derived from the CAM. Based on this, the network predicts a soft affinity score that quantifies the probability of region co-membership. The learned affinity matrix is then used to construct a random-walk transition matrix, which propagates and diffuses the initial CAM confidences. This process spreads high-confidence information along anatomically consistent paths to semantically related regions while suppressing spurious connections. The entire pipeline is fully differentiable, allowing GA-Net to be integrated with the backbone segmentation network for end-to-end joint training, as shown in Algorithm 1.

## Graph-based affinity learning

We formulate affinity refinement as probabilistic edge inference on a superpixel graph, where the goal is to estimate the posterior probability that two regions belong to the same anatomical structure. A GA-Net serves as a learned kernel producing edge posteriors under weak supervision from CAM, while random-walk propagation realizes structure-aware diffusion over a row-stochastic transition matrix. Below, we detail the derivations and design choices.

**Graph construction as a latent-variable model:** For any pair of superpixels connected in the graph, we define a binary latent variable $y \in \{0, 1\}$, where $y = 1$ indicates that the two regions belong to the same anatomical structure, and

**Algorithm 1 End-to-end training and inference pipeline of GA-Net.**

**Require:** Input image $I$, image-level label $b$

**Ensure:** Refined segmentation mask $M_{\text{aff}}$

1: **1. Feature extraction and graph construction**

2: Extract encoder features $\mathbf{f}^{\text{Enc}}$

3: Generate superpixel regions $\{P^{(i)}\}$ using SLIC

4: Construct graph $\mathcal{G} = (\mathcal{V}, \mathcal{E})$, including adjacency edges and kNN long-range edges

5: Initialize node features: $\mathbf{h}^{(0)} = [\mathbf{f}^{\text{Enc}} \oplus \mathbf{f}^{\text{app}} \oplus \mathbf{f}^{\text{coord}} \oplus a^{\text{init}}]$

6: Generate soft target affinity $w$ from CAM (Equation 2)

7: **2. GA-Net inference for edge affinity**

8: **for** $\ell = 0$ to $L-1$ **do**

9: **for** each edge $(i,j) \in \mathcal{E}$ **do**

10: Compute attention weight: $\alpha_{ij}^{(\ell)} = \text{softmax}\left(\mathbf{a}^{(\ell)\top}\left[\mathbf{W}^{(\ell)}\mathbf{h}_i^{(\ell)} \oplus \mathbf{W}^{(\ell)}\mathbf{h}_j^{(\ell)}\right]\right)$

11: **end for**

12: **for** each node $i \in \mathcal{V}$ **do**

13: Update node feature: $\mathbf{h}_i^{(\ell+1)} = \sigma\left(\sum_{j \in \mathcal{N}(i)} \alpha_{ij}^{(\ell)}\mathbf{W}^{(\ell)}\mathbf{h}_j^{(\ell)}\right)$

14: **end for**

15: **end for**

16: Compute edge posterior probability: $a_{ij}^{\text{opt}} = \sigma\left(\mathbf{U}^{\top}\left[\mathbf{h}_i^{(L)} \oplus \mathbf{h}_j^{(L)}\right]\right)$

17: **3. Loss computation and optimization**

18: Compute structural consistency loss:
$\mathcal{L}_{\text{struct}} = -\sum_{\mathcal{E}}\left[w \log a^{\text{opt}} + (1-w)\log(1 - a^{\text{opt}})\right]$

19: Compute smoothing regularization term: $\mathcal{L}_{\text{smooth}} = \sum_{\mathcal{E}} \omega \cdot \frac{1}{2}(a^{\text{opt}} - a^{\text{init}})^2$

20: Total loss: $\mathcal{L} = \mathcal{L}_{\text{struct}} + \lambda_1 \mathcal{L}_{\text{smooth}}$

21: Backpropagate and update GA-Net parameters

22: **4. Random-walk propagation (inference phase)**

23: Construct transition matrix: $T = D^{-1}(a^{\text{opt}})^{\eta}$, where $D_{ii} = \sum_j (a_{ij}^{\text{opt}})^{\eta}$

24: Propagate activations: $M_{\text{aff}} = T \cdot \text{vec}(M)$

25: **return** $M_{\text{aff}}$

$y = 0$ otherwise. Our goal is to estimate the posterior probability of this co-membership:

$$a^{\star} \triangleq \Pr(y = 1 \mid I) \tag{1}$$

where $I$ denotes the input image or volume, $y$ represents region co-membership, and $a^{\star}$ is the true posterior affinity between the two connected regions. Since direct supervision for $y$ is unavailable, a soft target is instead derived from the CAM overlap:

$$w = \frac{2|P^{(1)} \cap P^{(2)}|}{|P^{(1)}| + |P^{(2)}|} \in [0, 1] \tag{2}$$

where $P^{(1)}$ and $P^{(2)}$ are the pixel/voxel sets corresponding to the two regions, and $w$ represents the soft target affinity based on the overlap between these regions in the CAM. This soft target captures the expected co-membership under weak supervision. To reduce the dimensionality of the latent space, we apply SLIC (Simple Linear Iterative Clustering) [29] segmentation, which converts the pixel-level graph into a compact region graph $\mathcal{G} = (\mathcal{V}, \mathcal{E})$, where the feature vector for each node is initialized as:

$$\mathbf{h}^{(0)} = [\mathbf{f}^{\text{Enc}} \oplus \mathbf{f}^{\text{app}} \oplus \mathbf{f}^{\text{coord}} \oplus a^{\text{init}}] \tag{3}$$

where $\mathbf{f}^{\text{Enc}}$ represents the image encoder feature, $\mathbf{f}^{\text{app}}$ is the intensity appearance feature, $\mathbf{f}^{\text{coord}}$ is the spatial coordinate encoding (e.g., centroid), and $a^{\text{init}}$ is the initial affinity. The operator $\oplus$ denotes concatenation of these features.

**k-Nearest Neighbor (kNN) long-range edges:** Anatomical structures, such as elongated vessels, may be spatially separated but semantically similar. To account for this, we incorporate kNN edges in the feature space. This reduces the graph diameter and enables multi-hop consistency, while maintaining local planarity with adjacency edges. These long-range edges help propagate affinity information across regions that are anatomically related but spatially distant.

**GA-Net as a learned positive-definite Kernel:** A single GA-Net layer computes the normalized attention weights $\alpha^{(\ell)}$ using a learned similarity measure:

$$\alpha^{(\ell)} = \text{softmax}_{\mathcal{N}} \left( \mathbf{a}^{(\ell)\top} \left[ \mathbf{W}^{(\ell)} \mathbf{h}_{(1)}^{(\ell)} \oplus \mathbf{W}^{(\ell)} \mathbf{h}_{(2)}^{(\ell)} \right] \right) \tag{4}$$

$$\mathbf{h}^{(\ell+1)} = \sigma \left( \sum_{\mathcal{N}} \alpha^{(\ell)} \mathbf{W}^{(\ell)} \mathbf{h}^{(\ell)} \right) \tag{5}$$

where $\mathbf{W}^{(\ell)}$ is the learnable projection matrix, $\mathbf{a}^{(\ell)}$ is the attention vector, $\mathbf{h}_{(1)}^{(\ell)}$ and $\mathbf{h}_{(2)}^{(\ell)}$ are the features of the two connected nodes, $\alpha^{(\ell)}$ represents the normalized attention weight, $\mathcal{N}$ is the neighborhood set, and $\sigma(\cdot)$ is the activation function. The nodes indexed by (1) and (2) denote the two nodes in the edge, and no specific spatial dimensionality is implied.

**Attention vs. Fixed Kernels:** Medical textures in images are inherently non-stationary. By using learned attention, the model can adapt to the specific context of the image. This allows attention to down-weight misleading cues (such as noise or specularities) and up-weight important features like those from self-attention network near anatomical boundaries. Furthermore, the concatenation-based scoring method allows the model to capture cross-feature interactions that static kernels.

**From node embeddings to edge posteriors:** For each edge, the Bernoulli parameter [30] is predicted as:

$$a^{\text{opt}} = \sigma \left( \mathbf{U}^{\top} \left[ \mathbf{h}_{(1)}^{(L)} \oplus \mathbf{h}_{(2)}^{(L)} \right] \right) = \Pr_{\theta}(y = 1 \mid I). \tag{6}$$

This is equivalent to performing logistic regression on the final GA-Net embeddings, transforming the problem into a maximum-likelihood estimation task under the weak labels $w$.

**Loss as Maximum-A Posteriori (MAP) estimation:** The overall loss function combines a penalized negative log-likelihood term:

$$\mathcal{L} = \underbrace{\mathcal{L}_{\text{struct}}}_{\text{data term}} + \lambda_1 \underbrace{\mathcal{L}_{\text{smooth}}}_{\text{informative prior}}. \tag{7}$$

**Structural consistency (data term):** Since $w$ serves as a soft target for $y$, we use soft-label cross-entropy as the loss function:

$$\mathcal{L}_{\text{struct}} = -\sum_{\mathcal{E}} \left[ w \log a^{\text{opt}} + (1 - w) \log(1 - a^{\text{opt}}) \right]. \tag{8}$$

This is equivalent to minimizing the Kullback-Leibler divergence between the Bernoulli distributions Bern($w$) and Bern($a^{\text{opt}}$). The inclusion of both positive and negative terms in the loss helps reduce false-positive affinities, particularly in

noisy CAM scenarios. While the original formulation approximates a positive-class upper bound, the full soft binary cross-entropy (BCE) loss is more robust and less prone to overfitting. A practical note is that binary CAM can be replaced with a soft Dice/Jaccard form to ensure differentiability:

$$w = \frac{2\langle \hat{p}^{(1)}, \hat{p}^{(2)} \rangle}{\|\hat{p}^{(1)}\|_1 + \|\hat{p}^{(2)}\|_1 + \epsilon} \tag{9}$$

**Local smoothness (informative prior):** To stabilize the model, a Tikhonov prior [31] is applied to the initial affinity:

$$\mathcal{L}_{\text{smooth}} = \sum_{\mathcal{E}} \omega\, \phi\left(a^{\text{opt}} - a^{\text{init}}\right), \quad \phi(\Delta) = \frac{1}{2}\Delta^2 \tag{10}$$

Here, $\omega \in [0,1]$ is a confidence weight based on feature similarity or edge strength. This prior helps to anchor the weak supervision around the initial structure and accelerates convergence, while also preventing overly sharp or discontinuous affinity distributions. From a MAP perspective, it models the posterior as:

$$\Pr(a^{\text{opt}} \mid a^{\text{init}}) \sim \mathcal{N}(a^{\text{init}}, \sigma^2 \propto 1/\omega) \tag{11}$$

This smoothness prior is critical for ensuring numerical stability, particularly in random walk-based propagation, and contributes to the overall robustness of the model.

**Class-driven affinity pseudo label generation**

Let $\widetilde{\phi}$ denote the classification head, which comprises fully connected layers. We can derive the initial pseudo-label $Y_p$ for an input medical CT image $\mathbf{I} \in \mathbb{R}^{H \times W \times D}$ labeled with binary categories $\boldsymbol{b} \in \mathbb{R}^{1 \times 2}$ as: $Y_p = \widetilde{\phi}(\mathbf{I}, \boldsymbol{b})$, where every pixel contributes to the generation of the pseudo-label. Specifically, to generate the initial pseudo-label, $\widetilde{\phi}$ is trained using image-level labels with label smoothing cross-entropy loss [32], formulated as:

$$\mathcal{L}_{cls} = -\frac{1}{N}\sum_{i=1}^{N} \left[\tilde{y}_i \log(\hat{y}_i) + (1 - \tilde{y}_i)\log(1 - \hat{y}_i)\right], \tag{12}$$

where $\tilde{y}_i = (1 - \epsilon)y_i + \epsilon \frac{1}{K}$. $K$ represents the number of classes. This adjustment ensures a more balanced learning process by softening the targets, thereby improving the model's robustness against noisy labels.

**Initial pseudo label generation:** Deriving a reliable initial pseudo label $Y_{aff}$ is a crucial step in learning the semantic affinity $W_{aff}$. As illustrated in Fig 2, we extract $Y_{aff}$ from initial pseudo labels $Y_p$. Following [2,14,26], we utilize two thresholds, $\varphi_l$ and $\varphi_h$, where $0 < \varphi_l < \varphi_h < 1$, to categorize the $Y_p$ into three distinct regions: reliable foreground, background, and uncertain areas. Formally, given the CAM $M \in \mathbb{R}^{h \times w \times d \times C}$, the pseudo label $Y_p$ is constructed as follows:

$$Y_p^{i,j,k} = \begin{cases} \texttt{argmax}(M^{i,j,k,:}), & \text{if } \texttt{max}(M^{i,j,k,:}) \geq \varphi_h, \\ 0, & \text{if } \texttt{max}(M^{i,j,k,:}) \leq \varphi_l, \\ 1, & \text{otherwise,} \end{cases} \tag{13}$$

where 0 denotes the background and 1 denotes ignored regions. The $\texttt{argmax}(\cdot)$ function identifies the semantic class with the highest activation value.

**Affinity Converter (AC):** The Affinity Converter (AC) serves as a crucial component in our method, transforming raw feature representations into meaningful affinity relationships. For affinity elements $\boldsymbol{A}(k_1, k_2, k_3)$ in the three-dimensional

affinity matrix, it represents the affinity between the voxel at $(i_1, j_1, k_1)$ and $(i_2, j_2, k_2)$. The calculation of affinity can similarly be based on the vector representation of voxels $\boldsymbol{f_h}(i, j, k)$, as shown:

$$\boldsymbol{A}(k_1, k_2, k_3) =$$
$$\text{norm}\left(\frac{\boldsymbol{f_h}(i_1, j_1, k_1)^\top \boldsymbol{f_h}(i_2, j_2, k_2)}{\|\boldsymbol{f_h}(i_1, j_1, k_1)\|_2 \|\boldsymbol{f_h}(i_2, j_2, k_2)\|_2}\right) \tag{14}$$

**Focal affinity loss:** The generated $Y_{aff}$ will be used as a supervisory signal to guide the model in learning to predict an accurate semantic affinity representation $W_{aff}$. This supervision process is achieved by minimizing the discrepancy between the model's prediction $W_{aff}$ and the ground-truth label $Y_{aff}$. Specifically, we use a modified focal affinity loss term $\mathcal{L}_{aff}$, which takes the following form:

$$\mathcal{L}_{aff} = \left(\frac{1}{N^+} \sum_{(ijk,mno) \in \mathcal{K}^+} \log(1 + \exp(-W_{aff}^{ijk,mno}))\right)$$
$$+ \left(\frac{1}{N^-} \sum_{(ijk,mno) \in \mathcal{K}^-} \log(1 + \exp(W_{aff}^{ijk,mno}))\right) \tag{15}$$

where $N^+$ and $N^-$ denote the counts of $\mathcal{K}^+$ and $\mathcal{K}^-$ samples. $\mathcal{K}^+$ and $\mathcal{K}^-$ represent the sets of positive and negative samples in $Y_{aff}$, respectively.

**Region-wise affinity propagation:** To further refine the CAM using learned affinities, we introduce a Region-wise Affinity Propagation mechanism. This mechanism enables the propagation of activation scores from high-confidence regions to semantically similar but low-activation regions, leveraging the learned affinity matrix. Let the CAM volume be $M \in \mathbb{R}^{h \times w \times d \times C}$, and let the global semantic affinity matrix be $W_{aff} \in \mathbb{R}^{(hwd) \times (hwd)}$. We partition the CAM into $R$ overlapping local 3D regions, where each region $r \in \{1, 2, ..., R\}$ contains a subvolume $M^r$ and corresponding affinity submatrix $W_{aff}^r$. To perform affinity-based propagation within each region, we first construct a row-normalized transition matrix $T^r \in \mathbb{R}^{N_r \times N_r}$ based on the affinity submatrix:

$$T^r = (D^r)^{-1} \cdot (W_{aff}^r)^\eta, \quad D_{ii}^r = \sum_j (W_{aff}^r)_{ij}^\eta \tag{16}$$

where $\eta > 0$ controls the propagation strength, and $D^r$ is the diagonal normalization matrix to ensure each row of $T^r$ sums to 1. The refined activation $M_{aff}^r$ for region $r$ is computed by multiplying the transition matrix with the vectorized CAM:

$$M_{aff}^r = T^r \cdot \text{vec}(M^r) \tag{17}$$

To aggregate results across overlapping regions, we apply a distance-weighted merging strategy. Let $\mathcal{R}(i, j, k)$ denote all regions containing voxel $(i, j, k)$, and let $w_r(i, j, k)$ be the weight assigned to region $r$ based on the voxel's proximity to the region center $c_r$. The final propagated CAM is calculated as:

$$M_{aff}(i, j, k, c) = \frac{\sum\limits_{r \in \mathcal{R}(i,j,k)} w_r(i, j, k) \cdot M_{aff}^r(i - i_r, j - j_r, k - k_r, c)}{\sum\limits_{r \in \mathcal{R}(i,j,k)} w_r(i, j, k)} \tag{18}$$

## Basic supervised semantic segmentation

The initial pseudo labels $Y_p$ undergo a series of refinement processes, ultimately generating the pseudo labels $\Omega$ for the semantic segmentation task. In the pseudo-annotated semantic segmentation training, we adopt a smoothed cross-entropy loss $\mathcal{L}_{seg}$:

$$\mathcal{L}_{seg} = \frac{1}{|\Omega|} \sum_{y_i \in \Omega} \left[ -\sum_{k=1}^{K} \left( (1 - \epsilon) y_{i,k} + \epsilon \frac{1}{K} \right) \log(Y_{i,k}^p) \right]. \tag{19}$$

where $\epsilon = 0.15$ is a regularization term for label smoothing [32] to prevent the model from becoming overconfident. $K$ is the number of classes. $Y_i^p$ is the prediction from the model, $y_i$ is the one-hot label.

**Exponential moving average:** In the initial stages of training for WSSS, model predictions are often unstable and exhibit limited accuracy. To address this challenge, we propose an approach inspired by [11], wherein we employ a prediction ensembling technique known as Exponential Moving Average (EMA). This technique continuously accumulates and updates the EMA of predictions during training, effectively reducing the adverse impact of single prediction errors on the final results. Specifically, at each training iteration, the EMA combines the current model prediction $f(x; \theta)$ with the previous EMA prediction $y_{n-1}$, where the weights are determined by the smoothing factor $\delta$, following the update rule:

$$y_n = \delta f(x; \theta_i) + (1 - \delta) y_{n-1} \tag{20}$$

where $y_n$ represents the current EMA prediction, $f(x; \theta_i)$ is the current model prediction, $y_{n-1}$ is the previous EMA prediction, and $\delta$ is the smoothing factor. The predictions are averaged every $\gamma$ iterations, where $\gamma$ is the ensembling interval.

## Scribbled-based segmentation

To accurately segment small LC lesions, we propose a scribble-based segmentation approach. This method leverages scribble annotations to capture the complete shape and boundary information of LC target objects, thereby enhancing the model's ability to identify and process fine structures [12]. Unlike expert manual labels, these scribble annotations are indirectly generated by extracting and converting the contours of the pseudo labels $\Omega$ using the Lung Cavity Recognition Algorithm (LCRA) described in [2]. The pseudo labels provide approximate boundaries, which are then transformed into scribble annotations for improved LC segmentation accuracy. Specifically, we use a partial cross-entropy loss function $\mathcal{L}_{pce}$ as a sparse annotation supervision paradigm. This loss is defined as follows:

$$\mathcal{L}\_pce(\mathbf{I}, Y * s) = \\ -\frac{1}{|\Omega * s|} \sum *j \in \Omega * s \left[ Y\_s^j \log(f(\mathbf{I}; \theta)) + (1 - Y\_s^j) \log(1 - f(\mathbf{I}; \theta)) \right] \tag{21}$$

where $\mathbf{I}$ is the input CT image, $Y_s^j$ is the scribble label derived from the pseudo-label $\Omega$, and $\Omega_s$ is the set of scribbled pixels. $f(\mathbf{I}; \theta)$ represents the model's prediction.

**Network training.** The total loss of our method is a weighted combination of six separate loss functions: $\mathcal{L}_{cls}$, $\mathcal{L}_{struct}$, $\mathcal{L}_{smooth}$, $\mathcal{L}_{aff}$, $\mathcal{L}_{pce}$, $\mathcal{L}_{seg}$. This combination is expressed as:

$$\mathcal{L} = \mathcal{L}_{cls} + \lambda_1 \mathcal{L}_{struct} + \lambda_2 \mathcal{L}_{smooth} + \lambda_3 \mathcal{L}_{aff} + \lambda_4 \mathcal{L}_{pce} + \lambda_5 \mathcal{L}_{seg} \tag{22}$$

where $\lambda_1$, $\lambda_2$, $\lambda_3$, $\lambda_4$ and $\lambda_5$ are coefficients that balance the contributions of each loss function.

## Data and experiments

### Dataset

We conducted experiments on three publicly available TB chest CT datasets, covering both classification and segmentation tasks. Details of the datasets and data splitting strategies are summarized as follows. The classification criteria for lung cavity number and size across the three datasets are summarized in Table 1.

**ImageCLEF Tuberculosis (https://www.imageclef.org/2021/medical/tuberculosis):** This dataset [33] contains chest CT scans of 1,338 TB patients, each corresponding to a single patient and labeled with only one TB type. Some scans are accompanied by additional meta-information, which may vary by case. Among them, 917 images are designated for training and 421 for testing.

**TB Portals (https://tbportals.niaid.nih.gov/):** A global TB database and research platform developed by the National Institutes of Health (NIH) [34], containing 1,324 non-compound TB CT scans along with corresponding clinical data. We used the same categorical variables (LCs size and total cavity number), as shown in Table 1. In our study, 1,059 images were used for training and 265 for validation.

**DeepPulmoTB (https://github.com/SupCodeTech/DeepPulmoTB):** This dataset [7] includes 354 CT images with multi-class annotations for TB, including lung areas, LCs, and consolidation or lung cavity wall (C-LCW). Due to the limited dataset size, we adopted a 9:1 split strategy, using 318 images for training and 36 for validation.

### Label generation and refinement

**Image-level annotation generation.** Notably, in the Table 1, the ImageCLEF TB dataset does not provide annotations specific to LCs classification or segmentation, and the TB Portals dataset lacks pixel-level segmentation labels. To address this limitation, we collaborated with experts from the Department of Radiology at Universiti Putra Malaysia to construct reliable image-level supervision. The annotation process was conducted manually by three senior radiologists, each with over ten years of clinical experience in thoracic imaging. All annotations were performed using ITK-SNAP [35] under a unified and standardized protocol agreed upon by all annotators. Each patient CT volume was assigned image-level labels (e.g., 1-3 cavities and <25$mm$), which describe the shape characteristics of LC lesions and serve as the sole human supervision during the initial classification and CAM generation stages.

**Pseudo-label derivation and refinement.** Based on the image-level labels, a classification head was first trained to produce CAMs. These CAMs provide coarse localization cues but are inherently noisy and incomplete. To obtain initial pseudo pixel-level labels $Y_p$, we applied a dual-threshold strategy with thresholds $\varphi_l$ and $\varphi_h$, partitioning CAM responses into reliable foreground, background, and uncertain regions. Subsequently, structural affinity learning and region-wise affinity propagation were employed to refine these pseudo labels, resulting in progressively improved pseudo segmentation masks $\Omega$.

**Scribble annotation generation and supervision.** Unlike conventional approaches that rely on manually annotated scribbles, the scribble annotations in our method are generated automatically. Specifically, we extract the contours of the refined pseudo labels $\Omega$ and convert them into sparse scribble annotations using the Lung Cavity Recognition Algorithm

**Table 1.** The annotated classification of the lung cavity size and quantity in a patient's CT scan across three datasets. The "Class" column indicates the numerical index corresponding to each classification category.

| Total Cavern Num | Cavern Size | TB Portals | ImageCLEF | DeepPulmoTB | Class |
|---|---|---|---|---|---|
| No cavities | - | 318 | 343 | 125 | 1 |
| 1–3 cavities | <25mm | 305 | 317 | 86 | 2 |
| | >25mm | 314 | 284 | 46 | 3 |
| >3 cavities | <25mm | 225 | 205 | 52 | 4 |
| | >25mm | 162 | 189 | 45 | 5 |

(LCRA) proposed in [2]. These scribbles preserve essential shape and boundary information of lung cavity lesions while avoiding the need for additional expert annotation. The resulting scribble masks are then used to construct partial supervision masks $\Omega_s$, where only scribbled pixels contribute to the loss computation via a partial cross-entropy objective. This strategy enables effective learning from sparse supervision and further improves segmentation accuracy for small and irregular LCs lesions.

Overall, the proposed pipeline establishes a complete and coherent supervision hierarchy, starting from expert-provided image-level labels, progressing through affinity-refined pseudo-labels, and finally yielding scribble-based supervision masks for robust weakly supervised semantic segmentation.

### Data preprocessing and experimental setup

To ensure data consistency and model training stability, we applied a unified preprocessing pipeline to all datasets.

**Imaging modality and specifications**: All data were obtained from chest CT volumes acquired using standard clinical scanning protocols, in the original DICOM format. The specific parameters of CT images varied among cases: the in-plane resolution ranged from 0.6 × 0.6 mm to 1.0 × 1.0 mm, the slice thickness ranged from 1 mm to 5 mm, and each 3D volume contained approximately 100 to 400 consecutive slices. To eliminate spatial scale differences among data from different sources, we resampled all CT volumes to a uniform isotropic resolution of 1.0 × 1.0 × 1.0 mm³. Furthermore, the image gray values (Hounsfield Units, HU) were clipped to the window of [–1200,600] to focus on the typical density range of lung tissue, and then linearly normalized to the interval [0,1] for model input.

**Data augmentation**: During the training phase, we adopted an online data augmentation strategy based on MONAI to improve the generalization ability of the model. For each CT volume, we first loaded the image and its corresponding segmentation label, and uniformly converted the data into the channel-first format ( $C \times H \times W \times D$ ). Then, fixed-size three-dimensional patches were randomly cropped from the original volume, with the patch size set to 128×128×96 and random size variation disabled (random_size=False). This random 3D cropping was performed online at each iteration to increase the spatial diversity of training samples and reduce memory overhead. For intensity augmentation, we applied random Gaussian noise perturbation (RandGaussianNoise/RandGaussianNoised) only to the input images (img) to simulate imaging noise and intensity fluctuations caused by different scanning conditions, thereby enhancing the model's robustness to noise and intensity variations. It is important to emphasize that all spatial transformations (e.g., random cropping) were applied synchronously to both images and segmentation labels to ensure pixel-level alignment, while intensity-based transformations were applied only to images and not to segmentation labels, so as to avoid introducing label noise.

**Hardware and software environment**: In our experimental setup, all models were trained on a single NVIDIA RTX 3090Ti GPU with 24 GB of memory. During training, the batch size was set to 8, and the initial learning rate was $1 \times 10^{-4}$. Model optimization was performed using the AdamW optimizer, with a weight decay of 0.05 and momentum parameters $\beta_1 = 0.9$ and $\beta_2 = 0.95$. A warm-up strategy was applied at the beginning of training, during which the learning rate was linearly increased to $5.625 \times 10^{-5}$, followed by a cosine annealing learning rate scheduler to gradually decay the learning rate.

### Cross-validation protocol and statistical evaluation

To clarify the robustness of the statistical analysis and the evaluation protocol, we provide additional details on the cross-validation (CV) design and inferential testing strategy adopted in this study. For each dataset, a fixed held-out test set was first defined according to the protocol described in the Dataset section. The remaining data were used exclusively for training and validation. Within this training set, a 5-fold cross-validation strategy was employed for hyper-parameter selection and model configuration. Specifically, the training data were randomly partitioned into five non-overlapping folds at the patient level. In each CV iteration, four folds were used for model training, and the remaining fold was used for validation. This process was repeated five times, such that each fold served as the validation set once. The patient-level

data distribution for the 5-fold cross-validation across all datasets is summarized in Table 2. All hyperparameters, including those related to graph-based affinity learning, pseudo-label generation, and supervised segmentation, were selected based solely on the average validation performance across the five folds. Importantly, the held-out test set was never used during cross-validation, model selection, or hyper-parameter tuning. After determining the optimal configuration, the final model was retrained using the full training set and evaluated once on the held-out test set.

Moreover, to ensure robust and statistically sound comparisons, significance testing was performed on segmentation metrics (DSC, IoU) rather than on correlated training losses. Specifically, for each method, we obtained its prediction for every individual patient in the dataset. Since all methods were evaluated on the same set of patients, the DSC/IoU scores across patients formed paired samples for any pair of methods. We then applied the Wilcoxon signed-rank test to these paired patient-level scores to determine statistical significance. This approach provides a more reliable and powerful inference than using fold-aggregated scores (which would yield only 5 data points per method) or training losses. Table 3 summarizes the statistical evaluation protocol. All statistical analyses were performed using Python's SciPy library (version 1.10.1)

## Implementation details

To ensure reliable evaluation and stable model training, all experiments were conducted using a consistent experimental setup. The image encoder was fixed to ViT-B/16 as the backbone network across all experiments. Details of the cross-validation protocol, including data partitioning, hyper-parameter selection, and statistical evaluation, are provided in the previous section.

**Sensitivity analysis of key hyper-parameters for graph-based affinity learning.** To assess the robustness of the proposed method with respect to key hyper-parameters, we conducted a sensitivity analysis by varying the affinity thresholds, graph design parameters, and propagation settings around their default values. Specifically, the foreground and background thresholds ($\varphi_l, \varphi_h$) were perturbed within a reasonable range (e.g., ±10% of the default values), and the radius of the local 3D window was varied between (5,5,5) and (9,9,9). In addition, we evaluated alternative GA-Net depths in the range of 1 to 15 layers.

**1) Pseudo-label Generation Thresholds** ($\varphi_l, \varphi_h$)**:** These thresholds are used to partition reliable foreground, background, and uncertain regions from the initial CAM, and play a critical role in affinity learning. To evaluate the sensitivity of the proposed method to these parameters, we systematically examined different combinations of $\varphi_l$ and $\varphi_h$ within the ranges [0.25,0.35] and [0.40,0.50], respectively. Experimental results show that when ($\varphi_l, \varphi_h$) varies within these ranges,

**Table 2**. **Patient-level data distribution under 5-fold cross-validation.** For each dataset, a fixed test set is held out and excluded from cross-validation. Five-fold CV is performed only on the training pool at the patient level.

| Dataset | Fold | Training Patients | Validation Patients | Test Patients |
|---------|------|-------------------|---------------------|---------------|
| TB Portals | 1,2,3,4,5 | 847 | 212 | 265 |
| ImageCLEF | 1,2,3,4,5 | 734 | 183 | 421 |
| DeepPulmoTB | 1,2,3,4,5 | 254 | 64 | 36 |

**Table 3**. **Statistical evaluation protocol for segmentation performance.**

| Item | Description |
|------|-------------|
| Evaluation metric | DSC, IoU |
| Statistical unit | Patient-level |
| Test type | Paired Wilcoxon signed-rank test |
| Sample pairing | Same patient across methods |
| Significance level | $p < 0.05$ |
| Training loss testing | Not used for inferential statistics |

the final segmentation performance, measured by the DSC, remains on a consistently high and stable plateau with only minor fluctuations. The selected configuration ($\varphi_l = 0.32, \varphi_h = 0.46$) lies well within this high-performance region. Further statistical testing (patient-level paired Wilcoxon signed-rank test) confirms that there is no statistically significant difference in performance between the selected configuration ($\varphi_l = 0.32, \varphi_h = 0.46$) and other combinations within the interval (all comparisons $p > 0.05$). This indicates that the method is robust to small perturbations in the threshold parameters, and its effectiveness stems from the overall design rather than from precise tuning of specific threshold values.

**2) Local Affinity Window Radius:** Computing affinities within a local 3D window balances capturing sufficient context with computational efficiency and noise suppression. We tested cubic window sizes from 5 to 11 voxels per side. Performance peaked at a radius of 7 (i.e., a (7,7,7) window). Importantly, the performance degradation observed at radii of 5 or 9 compared to the peak performance was statistically non-significant ($p > 0.05$, patient-level paired Wilcoxon test). This suggests that the method does not critically depend on an exact window size and can achieve stable performance as long as it operates within a reasonable range of local context.

**3) GA-Net Depth on Graph-Based Affinity Learning:** For the graph-based affinity learning module, we systematically investigated the effect of GA-Net depth by varying the number of layers from 1 to 15. The DSC increased steadily as the network depth grew from 1 to 4 layers, indicating that deeper message passing enables more effective aggregation of contextual and structural information across superpixel regions. The performance reached its peak at four layers (as illustrated in Fig 3(a)), suggesting a favorable balance between local feature preservation and contextual integration. However, when the number of layers exceeds five, the segmentation performance exhibits a statistically significant decline (e.g., comparing 6 layers to 4 layers, $p < 0.05$). This performance degradation primarily stems from two aspects: first, excessive message passing may lead to over-smoothing in the graph domain, causing node representations from different anatomical regions to become increasingly similar, thereby weakening boundary discrimination; second, deeper graph propagation tends to diffuse affinity information into semantically unrelated or weakly connected regions, which is particularly detrimental when the weak supervision signals provided by CAMs are inherently noisy or incompletely localized. Based on this analysis, we select four layers as the default configuration for the GA-Net, as it consistently delivers strong and statistically superior performance while maintaining stable optimization behavior.

**Sensitivity analysis for the supervised segmentation module.** To rigorously evaluate the robustness of the exponential moving average (EMA) ensembling mechanism within the supervised segmentation module, we conducted a systematic sensitivity analysis on its two key hyper-parameters: the smoothing coefficient $\delta$ and the ensembling interval $\gamma$, in strict accordance with the statistical evaluation protocol summarized in Table 3. We tested $\delta$ across the values

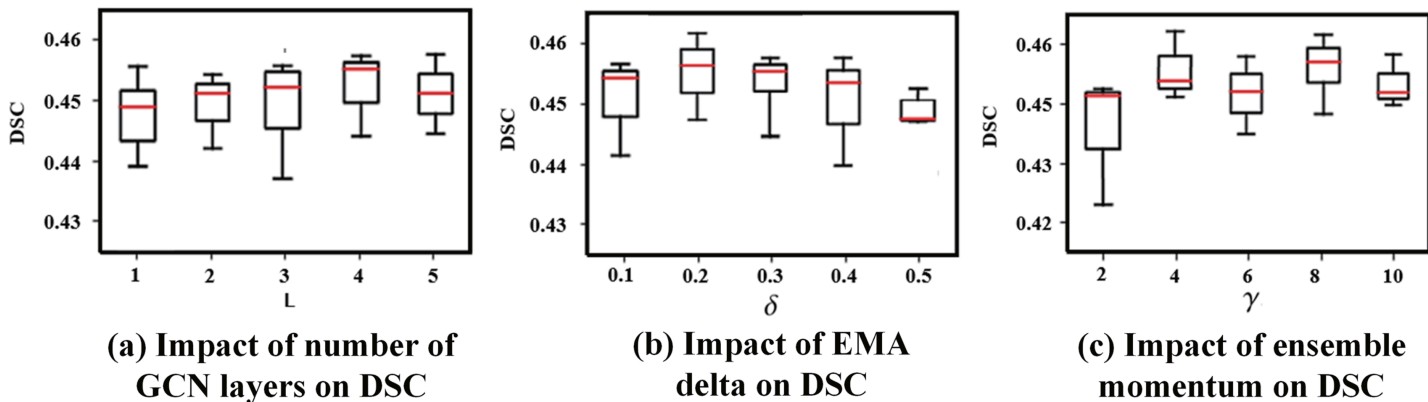

(a) Impact of number of GCN layers on DSC (b) Impact of EMA delta on DSC (c) Impact of ensemble momentum on DSC

**Fig 3**. Impact of parameter variations and module ablation on lung cavity segmentation under weak supervision.

{0.1, 0.2, 0.3, 0.4, 0.5, 0.7, 0.9} and $\gamma$ across {2, 4, 8, 10, 15, 20} using patient-level five-fold cross-validation. The segmentation performance, measured by the DSC on the held-out validation folds, is summarized in Fig 3(b) and 3(c). Statistical analysis using paired Wilcoxon signed-rank tests on patient-level DSC scores confirms the following robustness patterns:

1. **Smoothing coefficient $\delta$:** A broad plateau of high and statistically comparable performance was observed for $\delta \in$ [0.1, 0.3]. The performance achieved at the optimal setting ($\delta = 0.2$) was not significantly different from that at $\delta = 0.1$ or $\delta = 0.3$ (both $p > 0.05$). In contrast, extreme values ($\delta \leq 0.05$ or $\delta \geq 0.5$) resulted in a statistically significant degradation in performance ($p < 0.05$). Note: these extreme cases are not shown in the figure.
2. **Ensembling interval $\gamma$:** The segmentation performance remained consistently high and without statistically significant differences for $\gamma = 4$ and $\gamma = 8$ ($p > 0.05$). Significantly lower performance was observed only when the EMA updates were either too frequent ($\gamma = 2$) or too infrequent ($\gamma \geq 10$), with statistically significant differences ($p < 0.05$). Note: these extreme (> 10) cases are not shown in the figure.

The selected configuration ($\delta = 0.2$, $\gamma = 8$) lies well within these robust, high-performance intervals. This analysis, supported by formal statistical testing, demonstrates that the performance gains introduced by the EMA module stem from its core design principle–iterative prediction stabilization–rather than from meticulous fine-tuning of its hyper-parameters. Overall, the supervised segmentation module exhibits strong inherent stability across a reasonable range of parameter settings.

**Sensitivity analysis of loss weight parameters.** To rigorously assess the robustness of the proposed multi-task learning framework, we performed a systematic sensitivity analysis on the loss weighting coefficients $\lambda_1$ to $\lambda_5$ in Eq. (22), in accordance with the statistical evaluation protocol summarized in Table 3. A grid search was conducted within the bounded space $\lambda_i \in [0.1, 1.0]$ using patient-level cross-validation on the training set only, thereby ensuring that no test data were involved in hyper-parameter selection.

The analysis was designed to examine whether the segmentation performance was critically dependent on a specific combination of loss weights. To this end, we compared the DSC achieved by the final selected configuration ($\lambda_1 = 0.6, \lambda_2 = 0.5, \lambda_3 = 0.5, \lambda_4 = 0.7, \lambda_5 = 0.5$) with multiple neighboring configurations, in which each individual weight was independently perturbed by $\pm 0.2$ while keeping the remaining coefficients fixed. Statistical analysis based on a series of patient-level paired Wilcoxon signed-rank tests revealed no statistically significant performance differences between the chosen configuration and its immediate variants (all $p > 0.05$). These results indicate the presence of a broad plateau of stable performance rather than a single, sharply defined optimum.

## Ablation studies

To evaluate the contribution of different loss components in the method, we performed ablation experiments with the NIH "TB Portals" [34], ImageCLEF [33], and DeepPulmoTB dataset [7]. The ablation experiment design is shown in Table 4. The baseline method, M-5, incorporates all the loss components and serves as the reference for comparison. The results are summarized in Table 5, which presents the performance of various modules on segmentation and classification tasks. Fig 4 shows a performance comparison between conventional affinity methods and graph-based affinity methods. The corresponding decision-making process and recognition performance of CAM for the ablation experiments M-1 to M-5 are visualized in Fig 5.

**M-1 (Excluding $\mathcal{L}_{\text{struct}}$):** In this experiment, we removed the structural consistency loss $\mathcal{L}_{\text{struct}}$ to analyze its impact on model performance.Patient-level paired Wilcoxon signed-rank tests confirmed a statistically significant performance decline in both segmentation (DSC: $p < 0.01$) and classification accuracy compared to the full model M-5. This indicates that the structural consistency loss is essential for maintaining prediction coherence, particularly in regions with ambiguous boundaries. The CAM visualization in Fig 5 highlights the model's inability to maintain consistent boundaries, with

**Table 4**. Design of ablation study modules with different loss components used in our weakly supervised method.

| ID | $\mathcal{L}_{cls}$ | $\mathcal{L}_{seg}$ | $\mathcal{L}_{struct}$ | $\mathcal{L}_{aff}$ | $\mathcal{L}_{smooth}$ | $\mathcal{L}_{pce}$ |
|---|---|---|---|---|---|---|
| M-1 | ✓ | ✓ | × | ✓ | ✓ | ✓ |
| M-2 | ✓ | ✓ | ✓ | × | ✓ | ✓ |
| M-3 | ✓ | ✓ | ✓ | ✓ | × | ✓ |
| M-4 | ✓ | ✓ | ✓ | ✓ | ✓ | × |
| M-5 | ✓ | ✓ | ✓ | ✓ | ✓ | ✓ |

**Table 5**. The contribution of various ablation components to the lung cavity semantic segmentation task in the proposed weakly supervised method.

| Dataset | ID | Segmentation ($\pm$ SD) | | | Classification | | | | |
|---|---|---|---|---|---|---|---|---|---|
| | | 95HD | IoU | DSC | ACC | REC | PRE | FPR | F1 |
| TB Portals | M-1 | 35.12 $\pm$ 4.52 | 0.226 $\pm$ 0.135 | 0.403 $\pm$ 0.141 | 0.705 | 0.762 | 0.775 | 0.466 | 0.768 |
| | M-2 | 33.47 $\pm$ 5.95 | 0.243 $\pm$ 0.127 | 0.429 $\pm$ 0.122 | 0.713 | 0.769 | 0.781 | 0.445 | 0.775 |
| | M-3 | 31.25 $\pm$ 3.65 | 0.254 $\pm$ 0.142 | 0.437 $\pm$ 0.138 | 0.721 | 0.778 | 0.789 | 0.437 | 0.783 |
| | M-4 | 29.73 $\pm$ 5.72 | 0.262 $\pm$ 0.126 | 0.442 $\pm$ 0.122 | 0.724 | 0.790 | 0.791 | 0.428 | 0.791 |
| | M-5 | 27.94 $\pm$ 4.63 | 0.273 $\pm$ 0.128 | 0.454 $\pm$ 0.121 | 0.731 | 0.789 | 0.802 | 0.409 | 0.795 |
| ImageCLEF | M-1 | 36.42 $\pm$ 4.29 | 0.217 $\pm$ 0.142 | 0.386 $\pm$ 0.137 | 0.677 | 0.732 | 0.745 | 0.483 | 0.738 |
| | M-2 | 34.71 $\pm$ 6.43 | 0.233 $\pm$ 0.117 | 0.411 $\pm$ 0.126 | 0.685 | 0.739 | 0.750 | 0.462 | 0.744 |
| | M-3 | 32.48 $\pm$ 3.39 | 0.244 $\pm$ 0.152 | 0.419 $\pm$ 0.135 | 0.692 | 0.748 | 0.757 | 0.454 | 0.752 |
| | M-4 | 30.89 $\pm$ 6.06 | 0.251 $\pm$ 0.118 | 0.424 $\pm$ 0.128 | 0.696 | 0.757 | 0.760 | 0.445 | 0.760 |
| | M-5 | 29.07 $\pm$ 4.44 | 0.262 $\pm$ 0.132 | 0.436 $\pm$ 0.115 | 0.703 | 0.759 | 0.771 | 0.426 | 0.764 |
| DeepPulmoTB | M-1 | 37.93 $\pm$ 4.75 | 0.208 $\pm$ 0.128 | 0.371 $\pm$ 0.151 | 0.649 | 0.702 | 0.714 | 0.503 | 0.707 |
| | M-2 | 36.15 $\pm$ 5.47 | 0.224 $\pm$ 0.137 | 0.395 $\pm$ 0.115 | 0.657 | 0.709 | 0.719 | 0.481 | 0.714 |
| | M-3 | 33.75 $\pm$ 3.91 | 0.234 $\pm$ 0.132 | 0.402 $\pm$ 0.141 | 0.664 | 0.716 | 0.726 | 0.472 | 0.721 |
| | M-4 | 32.11 $\pm$ 5.38 | 0.241 $\pm$ 0.134 | 0.407 $\pm$ 0.117 | 0.667 | 0.726 | 0.728 | 0.462 | 0.728 |
| | M-5 | 30.18 $\pm$ 4.77 | 0.251 $\pm$ 0.124 | 0.418 $\pm$ 0.126 | 0.673 | 0.727 | 0.737 | 0.442 | 0.732 |

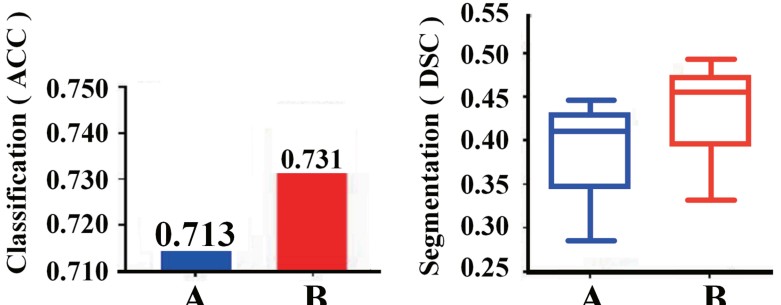

**Fig 4**. **Performance of conventional affinity and our graph affinity method.** A: conventional affinity method. B: graph-based affinity methods.

regions appearing fragmented and misclassified, reinforcing that $\mathcal{L}_{struct}$ significantly enhances structural integrity across anatomical regions.

**M-2 (Excluding $\mathcal{L}_{aff}$):** The exclusion of the affinity loss $\mathcal{L}_{aff}$ in M-2 further demonstrates the importance of capturing semantic affinities between regions. Statistical analysis showed a significant decrease in segmentation performance (DSC: $p < 0.01$), with boundaries becoming less precise. The CAM in Fig 5 illustrates that without $\mathcal{L}_{aff}$, the model fails to effectively capture relationships between anatomically connected regions, confirming its crucial role in segmenting connected structures.

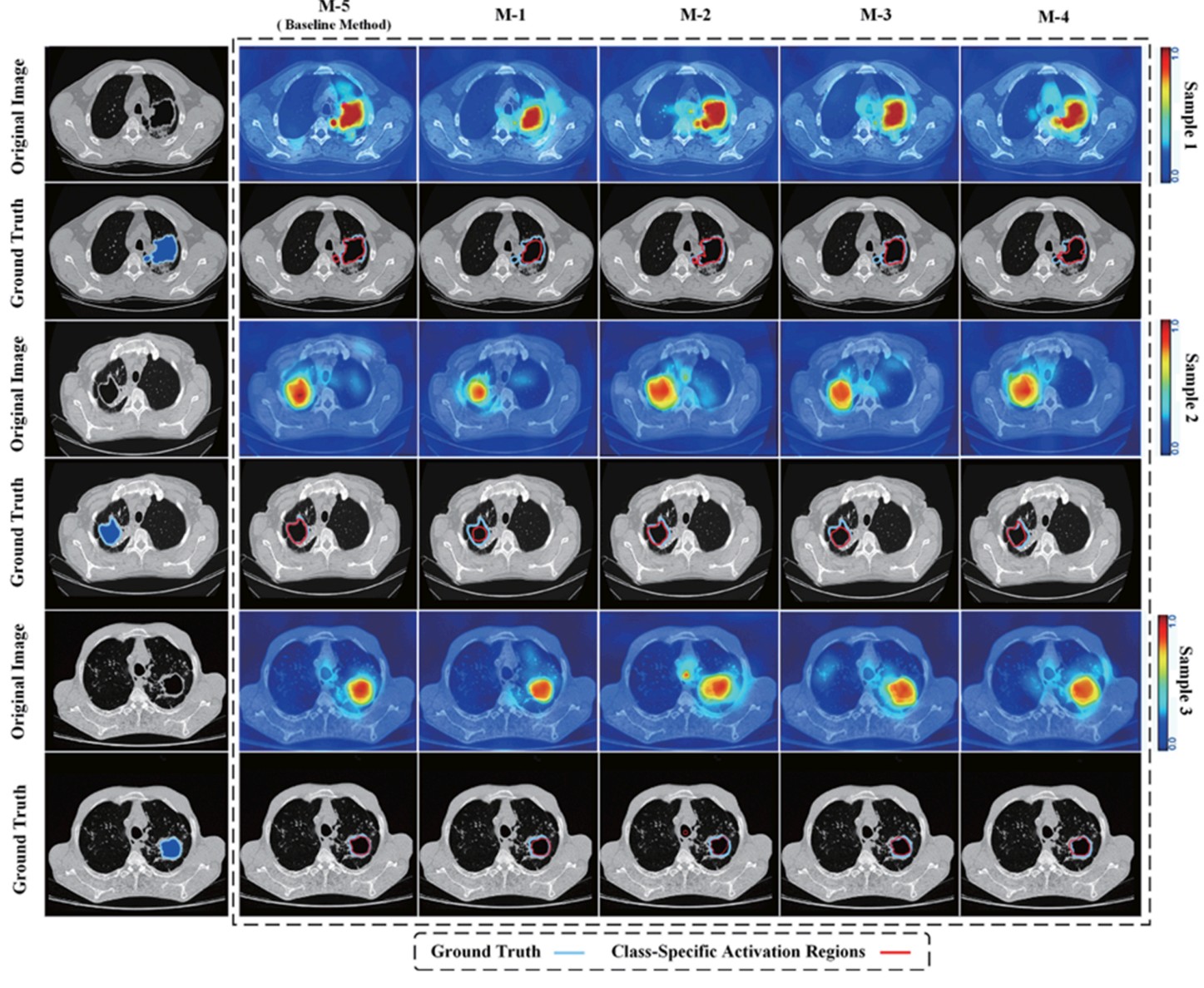

**Fig 5**. **Recognition results of class activation mapping for M-1 to M-5 ablation methods.**

**M-3 (Excluding $\mathcal{L}_{smooth}$):** Removing the smoothness loss $\mathcal{L}_{smooth}$ in M-3 resulted in a less stable model, especially under noisy or inconsistent supervision. Although the average segmentation accuracy remained relatively high, the performance variation across patients increased significantly, and the model outputs became more erratic ($p < 0.05$ for increased variance). The CAM in Fig 5 visualizations exhibited more jagged boundaries, indicating that smoothness regularization plays an important role in suppressing noise and stabilizing predictions.

**M-4 (Excluding $\mathcal{L}_{pce}$):** The M-4 experiment, which excluded the partial cross-entropy loss ($\mathcal{L}_{pce}$), showed a statistically significant but modest performance decrease compared to M-5 (DSC: $p < 0.05$). These results suggest that while the scribble-based module provides beneficial fine-grained boundary information, its absence particularly affects small

and irregular lesions. As shown in Fig 5, the CAM for M-4 exhibits less refined segmentation of LCs with weaker boundary definition, emphasizing the contribution of $\mathcal{L}_{\text{pce}}$ to capturing fine structural details.

**M-5 (Full Model with All Loss Components):** The full model (M-5) incorporates all proposed loss components and achieves the best performance across all evaluation metrics.Statistical comparisons against all ablated variants confirmed its significant superiority (all $p < 0.01$), demonstrating the combined effectiveness of the proposed loss terms. In addition, the CAM visualization in Fig 5 shows well-defined boundaries and accurate anatomical localization. Fig 4 demonstrates that the proposed graph-based affinity algorithm significantly outperforms conventional affinity propagation methods.

Overall, the ablation experiments, supported by rigorous patient-level statistical testing, confirm that each loss component contributes positively to the proposed weakly supervised framework. Specifically, the structural consistency, affinity, and smoothness losses play statistically verifiable roles in preserving boundary coherence, semantic connectivity, and prediction stability, while the scribble-based segmentation module significantly enhances fine-detail recognition for small or irregular lesions.

Moreover, Fig 6 presents a comparative analysis of the training process across five ablation methods (M-1 to M-5). Subfigures (a)–(e) correspond to the loss curves of each method, where the $x$-axis represents training epochs and the $y$-axis denotes the loss value. The experiments demonstrate that the training and validation losses for all methods show a convergent trend as epochs increase, and the gap between the validation and training curves remains below 0.15, indicating no overfitting in the models. The magnified views (insets) reveal that the loss values of M-5 in the final stabilization phase (last 50 epochs)—with training loss $0.552 \pm 0.03$ and validation loss $0.467 \pm 0.05$ (mean $\pm$ standard deviation)—are numerically lower than those of other methods. Quantitative analysis (Fig 6) shows that M-5 attains a DSC score of 0.457, outperforming methods M-4 to M-1 by 0.06–0.28.

Fig 7 presents a comparative analysis of classification performance across five ablation methods (M-1 to M-5) using confusion matrices, where diagonal elements represent correctly classified samples and off-diagonal elements indicate

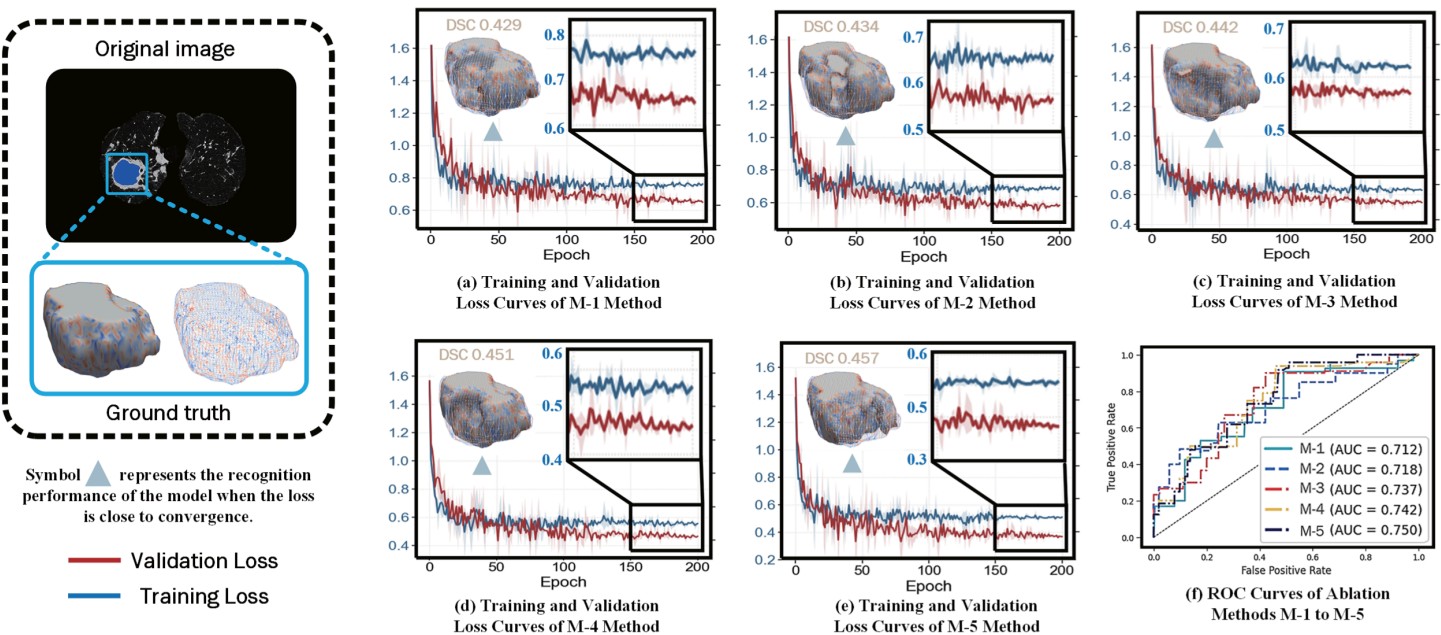

**Fig 6**. Training and validation losses of methods M-1 to M-5 and ROC curve performance for lung cavity attribute classification.

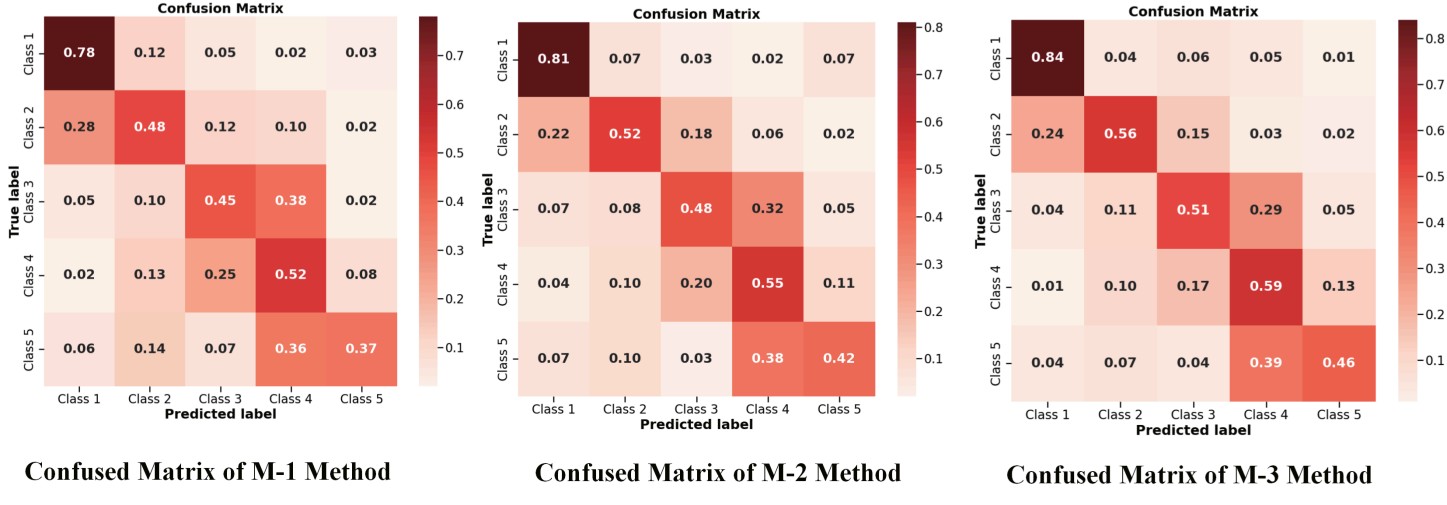

| Confused Matrix of M-1 Method | Confused Matrix of M-2 Method | Confused Matrix of M-3 Method |

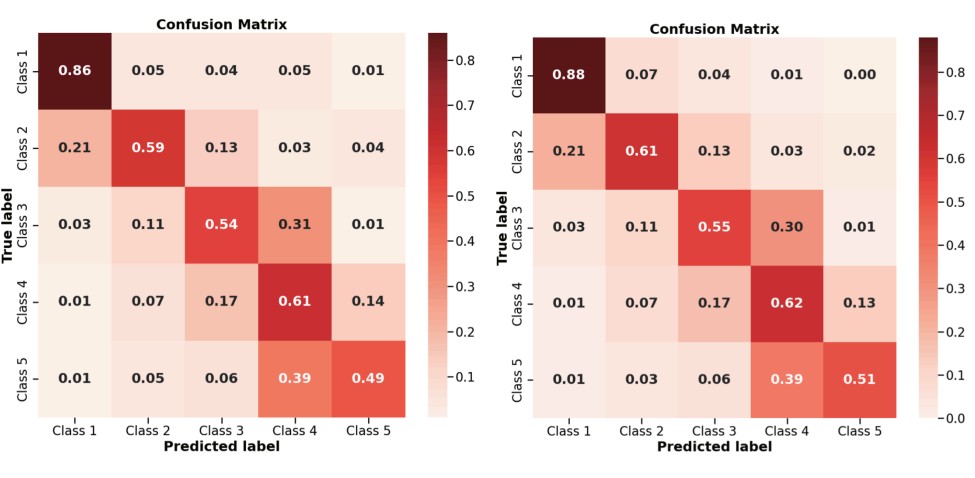

| Confused Matrix of M-4 Method | Confused Matrix of M-5 Method |

**Fig 7**. Confusion matrices for M-1 to M-5 ablation classification experiments.

misclassifications. The experimental results demonstrate that: 1) LCs categories with similar morphological characteristics (e.g., "no cavity" vs. "small cavities <25 mm", "no cavity" vs. "1–3 cavities") exhibit significant cross-confusion; 2) The baseline method M-5 achieves optimal classification performance. Fig 6(f) visualizes the ROC curves of each ablation method (M-1 to M-5) in the LCs attribute classification task. The figure shows that the baseline method incorporating all loss functions (M-5) achieved the best AUC in the LCs classification task, with a value of 0.750. These results validate the superiority of our proposed method in TB cavity attribute classification.

## Comparison to state-of-the-art

To comprehensively evaluate the performance of our method, we trained and compared several existing medical imaging WSSS methods on the LCs segmentation task, as summarized in Tables 6 and 7. Table 6 presents the benchmark configuration and comparative methods, including their supervision types and backbone architectures. The compared methods include recent state-of-the-art approaches, each designed with varying strategies for weakly supervised segmentation. Notably, some of these methods were originally developed in a 2D manner. To ensure a fair and consistent comparison

**Table 6**. **Benchmark configuration and fairness comparison across weakly supervised semantic segmentation methods.** All methods are re-trained under identical settings with a patch size of 128 × 128 × 96 and optimized using AdamW.

| Method | Venue | Supervision Type | Backbone |
|---|---|---|---|
| [36] | WACV | Point | VGG16 FCN8 |
| [19] | PR | Bounding Box | Faster RCNN |
| [37] | TMI | Scribble/Bounding Box/Point | Vanilla UNet |
| [15] | CVPR | Image-level | ResNet-38 |
| [24] | TMI | Bounding Box | UNet |
| [38] | TMI | Scribble | PRNet |
| [39] | TMI | Image-level | SAM's ViT |
| Ours | | Image-level | ViT-B/16 |

**Table 7**. **Quantitative results of various 3D medical weakly supervised (lung cavity) semantic segmentation methods.**

| Methods | TB Portals | | | ImageCLEF | | | DeepPulmoTB | | |
|---|---|---|---|---|---|---|---|---|---|
| | 95HD | IoU | DSC | 95HD | IoU | DSC | 95HD | IoU | DSC |
| [36] | 38.49 ± 4.26 | 0.224 ± 0.121 | 0.368 ± 0.123 | 39.03 ± 4.08 | 0.215 ± 0.130 | 0.353 ± 0.118 | 39.97 ± 4.59 | 0.206 ± 0.116 | 0.339 ± 0.113 |
| [19] | 36.48 ± 5.61 | 0.255 ± 0.112 | 0.409 ± 0.118 | 37.94 ± 5.33 | 0.245 ± 0.120 | 0.393 ± 0.113 | 38.40 ± 6.05 | 0.235 ± 0.103 | 0.376 ± 0.109 |
| [37] | 35.16 ± 6.37 | 0.238 ± 0.118 | 0.387 ± 0.112 | 36.57 ± 6.05 | 0.228 ± 0.113 | 0.372 ± 0.107 | 37.17 ± 6.88 | 0.219 ± 0.109 | 0.356 ± 0.103 |
| [15] | 33.12 ± 5.92 | 0.243 ± 0.114 | 0.414 ± 0.117 | 35.48 ± 5.62 | 0.233 ± 0.110 | 0.397 ± 0.112 | 35.85 ± 6.39 | 0.224 ± 0.105 | 0.381 ± 0.108 |
| [24] | 32.75 ± 5.78 | 0.248 ± 0.116 | 0.427 ± 0.119 | 34.12 ± 5.49 | 0.238 ± 0.117 | 0.410 ± 0.114 | 34.48 ± 6.25 | 0.229 ± 0.107 | 0.393 ± 0.110 |
| [38] | 30.89 ± 5.65 | 0.261 ± 0.117 | 0.442 ± 0.120 | 32.25 ± 5.37 | 0.251 ± 0.112 | 0.424 ± 0.115 | 32.62 ± 6.12 | 0.240 ± 0.108 | 0.407 ± 0.111 |
| [39] | 28.95 ± 5.43 | 0.266 ± 0.119 | 0.448 ± 0.122 | 30.32 ± 5.16 | 0.255 ± 0.114 | 0.430 ± 0.117 | 31.69 ± 5.89 | 0.245 ± 0.109 | 0.412 ± 0.113 |
| Ours | **27.94 ± 4.63** | **0.273 ± 0.128** | **0.454 ± 0.121** | **29.07 ± 4.44** | **0.262 ± 0.132** | **0.436 ± 0.115** | **30.18 ± 4.77** | **0.251 ± 0.124** | **0.418 ± 0.126** |

with our proposed 3D method, we extended these 2D approaches to their 3D counterparts by applying them slice-wise across all dimensions and aggregating the volumetric outputs accordingly. This adaptation enables the evaluation of all methods under a unified 3D experimental setting that aligns with the nature of our input data and task requirements. The quantitative performance comparisons across three public datasets are reported in Table 7.

According to the statistical evaluation protocol defined in Table 3, we analyzed the results using the paired Wilcoxon signed-rank test (significance level $p < 0.05$) with the patient as the statistical unit. In multiple comparisons with the best baseline method, our proposed method achieved the best performance across all three key evaluation metrics–95HD, IoU, and DSC–and demonstrated statistically significant advantages ($p < 0.05$) on all three datasets (TB Portals, Image-CLEF, and DeepPulmoTB). Specifically, on the TB Portals dataset, our method achieved results of 27.94 ± 4.63 (95HD), 0.273 ± 0.128 (IoU), and 0.454 ± 0.121 (DSC), significantly outperforming all existing methods. On the ImageCLEF dataset, the results were 29.07 ± 4.44 (95HD), 0.262 ± 0.132 (IoU), and 0.436 ± 0.115 (DSC), again showing notable improvement over previous methods. On the DeepPulmoTB dataset, our method obtained 30.18 ± 4.77 (95HD), 0.251 ± 0.124 (IoU), and 0.418 ± 0.126 (DSC), maintaining its performance advantage.

In comparison to existing state-of-the-art methods, such as [36], [19], and [15], which show a consistent gap in performance (especially in terms of DSC and IoU), our method stands out in its ability to capture both detailed and large-scale anatomical features. This is reflected in the sharp improvements in the DSC scores across all datasets, with our method consistently achieving the highest scores. Furthermore, Fig 8 illustrates the segmentation results of various methods,

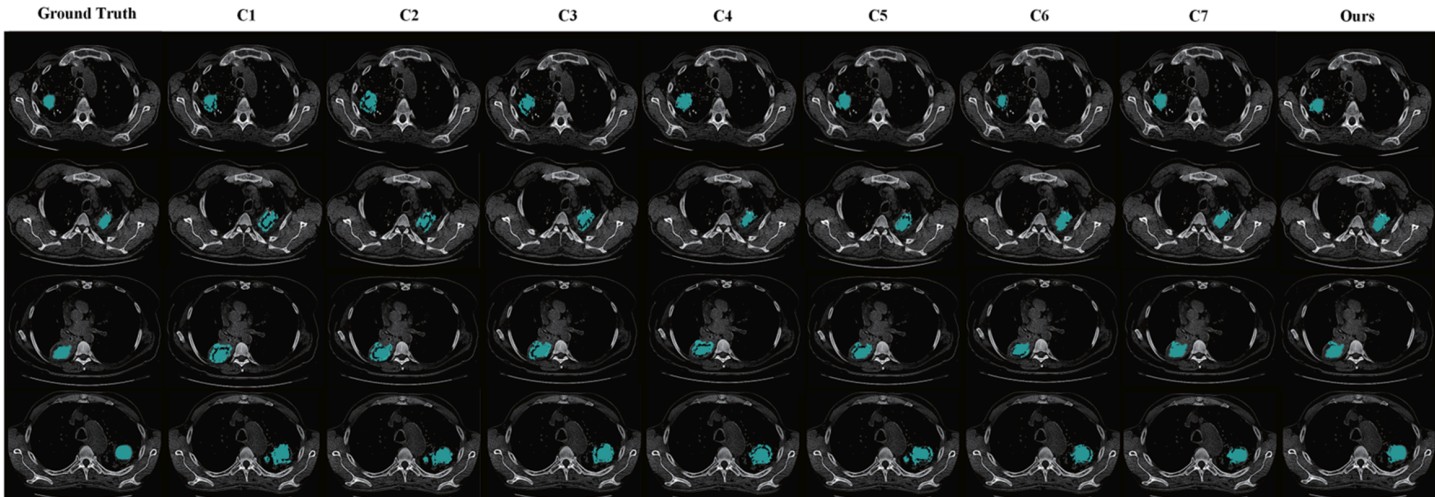

**Fig 8**. **Semantic segmentation results of lung cavities using various methods.** C1 - C7 sequentially represent the methods from [36], [19], [37], [15], [24], [38], and [39].

where it can be observed that our method achieves higher alignment with the ground truth compared to other approaches. The improved performance can be attributed to the integration of novel components, such as the GA-Net-based affinity learning and region-wise affinity propagation, which allow our model to better capture the relationships between anatomical structures and refine the segmentation boundaries.

Our method's superior performance suggests that the proposed method effectively addresses the challenges in weakly supervised 3D medical image segmentation, particularly for LC segmentation. The results indicate that our approach is more capable of handling complex anatomical structures and noisy data, thereby providing a more reliable and accurate segmentation solution compared to the state-of-the-art methods.

### Subgroup analysis by lesion characteristics

To better analyze the strengths and limitations of the proposed method, we further conducted qualitative and quantitative subgroup analyses based on lesion size, quantity, morphology, and disease severity. These subgroup definitions are derived from the cavity size and number annotations summarized in Table 1.

**Lesion size and quantity.** Quantitative analysis shows that the proposed method achieves more pronounced improvements over baseline methods in cases with multiple cavities (more than one cavity) and larger cavity sizes ($>25mm$), as summarized in Table 8. Within these subgroups, graph-based affinity learning can effectively bridge spatial distances and propagate semantic information between regions associated with relevant anatomical structures, thereby improving the completeness of the segmented regions and reducing fragmentation. In contrast, segmentation becomes more challenging for very small cavities ($<25mm$) or single-cavity cases, where partial volume effects and low contrast often lead to under-segmentation. Nevertheless, even under such conditions, our method consistently maintains a relative performance advantage over other approaches across all lesion-size-based subgroups.

**Lesion location.** Qualitative inspection indicates that cavities located in central lung regions or near major bronchovascular structures are segmented more reliably than those adjacent to pleural boundaries, as summarized in Table 9. Peripheral cavities often exhibit ambiguous boundaries and limited contextual support, which may reduce affinity propagation effectiveness. These failure cases are primarily associated with weak CAM localization rather than errors in the affinity learning module itself.

**Table 8. Subgroup analysis by cavity size and quantity.** Average DSC (%) across different lesion subgroups.

| Lesion Subgroup | Baseline | +Affinity | +Scribble | Ours |
|---|---|---|---|---|
| No cavity | – | – | – | – |
| 1–3 cavities (<25 mm) | 31.2 | 34.5 | 36.1 | **38.4** |
| 1–3 cavities (>25 mm) | 35.6 | 39.8 | 41.0 | **44.7** |
| >3 cavities (<25 mm) | 33.4 | 37.9 | 39.2 | **42.6** |
| >3 cavities (>25 mm) | 37.1 | 42.3 | 44.0 | **47.9** |

**Table 9. Qualitative performance analysis by lesion location.**

| Lesion Location | Performance Trend | Common Failure Mode |
|---|---|---|
| Central lung regions | Strong | Minor boundary leakage |
| Peri-bronchial | Strong | Partial under-segmentation |
| Peripheral (pleural-adjacent) | Moderate | Boundary ambiguity |
| Subpleural thin cavities | Weak | Missed or fragmented regions |

**Lesion morphology.** From a morphological perspective, the proposed method performs particularly well on cavities with irregular shapes or fragmented appearances, as summarized in Table 10. By modeling region-level affinities on a superpixel graph, GA-Net is able to capture long-range structural consistency that pixel-level methods often miss. However, thin-walled cavities or cavities with extremely complex internal structures may still suffer from boundary leakage, especially under weak supervision.

**Disease severity.** When stratifying cases by disease severity using cavity number as a proxy, we observe that the proposed method yields larger relative gains in moderate-to-severe cases (multiple cavities) compared to mild cases (single or no cavity), consistent with the quantitative trends reported in Table 8. This suggests that the proposed approach is particularly effective in scenarios where richer structural context is available, while extremely mild cases remain challenging due to limited lesion evidence.

Overall, these subgroup analyses demonstrate that the proposed method consistently improves segmentation performance across a wide range of lesion characteristics, while also highlighting specific scenarios–such as very small or peripheral cavities–where further improvements are needed.

## Computational complexity analysis

To assess the practical feasibility of GA-Net for clinical deployment, we analyze its computational complexity in terms of both theoretical operations and actual runtime performance.

**Theoretical complexity:** Let $N = |\mathcal{V}|$ be the number of superpixel nodes and $E = |\mathcal{E}|$ the number of edges in the constructed graph. Each GA-Net layer performs two main operations: (1) edge-wise attention scoring, and (2) node feature aggregation. The attention scoring step computes a scalar for each edge via a learnable projection followed by a dot product, requiring $O(E \cdot d)$ operations where $d$ is the feature dimension. The aggregation step then updates each node by a weighted sum of its neighbors, incurring $O(E \cdot d)$ operations as well. Thus, for $L$ layers, the overall complexity is $O(L \cdot E \cdot d)$.

**Table 10. Subgroup analysis by cavity morphology.**

| Morphology Type | Relative Performance | Remarks |
|---|---|---|
| Regular, round cavities | High | Clear boundaries |
| Irregular cavities | High | Benefited from graph affinity |
| Fragmented cavities | Moderate–High | Improved connectivity |
| Thin-walled cavities | Moderate | Boundary leakage observed |
| Complex internal structures | Moderate | Weak supervision limits accuracy |

In practice, we keep the graph sparse: adjacency edges are limited to local connectivity (typically 4–8 neighbors per node in 2D, 6–26 in 3D), and long-range kNN edges add a small constant factor (we use $k = 10$). Hence, $E = O(N)$, and the complexity scales linearly with the number of nodes, i.e., $O(L \cdot N \cdot d)$.

**Empirical runtime and memory:** We measured the forward-pass time and GPU memory consumption on a representative 3D CT volume (size $128 \times 128 \times 96$) using an NVIDIA RTX 3090Ti. After SLIC superpixel segmentation (which is a pre-processing step and can be efficiently parallelized), the graph contains $N \approx 2,000$ nodes and $E \approx 15,000$ edges. With a 4-layer GA-Net ($d = 256$), the affinity inference takes $\approx 42$ ms per volume, which translates to $\approx 0.66$ ms per slice (96 slices). The peak GPU memory footprint is $\approx 1.2$ GB, dominated by the node feature matrices and the attention weights. For comparison, a standard 3D U-Net baseline with similar feature dimensions requires $\approx 120$ ms per volume and $\approx 3.5$ GB of memory. Thus, GA-Net is $\approx 2.9\times$ faster and uses $\approx 2.9\times$ less memory than the U-Net baseline, making it highly suitable for real-time or near-real-time clinical scenarios.

**Comparison with alternative affinity methods:** We also compare against two graph-based affinity learning alternatives: (1) a non-parametric method that computes affinities via fixed cosine similarity on encoder features, and (2) a Graph Convolutional Network (GCN) with fixed edge weights. The non-parametric method is slightly faster ($\approx 35$ ms) but yields significantly lower segmentation accuracy (mIoU drops by 8.2% on our validation set). The GCN baseline has similar runtime to GA-Net ($\approx 45$ ms) but lacks the adaptive attention mechanism, leading to a 4.5 % mIoU reduction. Hence, GA-Net achieves a favorable trade-off between efficiency and accuracy.

**Clinical deployment considerations:** The linear scaling with respect to the number of nodes ensures that the model remains efficient even for high-resolution volumes. In a typical clinical workflow, the entire pipeline–including superpixel generation, GA-Net inference, and random-walk refinement–runs in under 200 ms per volume on a single GPU, meeting the real-time requirements for interactive segmentation tools. Furthermore, the model can be easily optimized via TensorRT or ONNX runtime for further speedup on embedded devices.

In summary, GA-Net introduces minimal computational overhead compared to conventional segmentation networks, while providing the structured reasoning necessary for robust affinity learning. Its low latency and memory footprint make it a practical choice for clinical deployment, where both accuracy and efficiency are critical.

## Discussion

The method proposed in this study represents a meaningful advance in the task of weakly supervised semantic segmentation of lung cavities in 3D medical images. Experimental results demonstrate that, compared to existing baseline methods, our model achieves significant relative performance improvements across multiple datasets (e.g., an average relative improvement of approximately 5% in DSC). This improvement validates the effectiveness of the proposed graph affinity learning and scribble-guided optimization mechanisms in leveraging weak supervision signals.

We note that, while the relative improvements are significant, there remains room for enhancement in the absolute performance levels of segmentation for this task (e.g., an average DSC of approximately 45.4% and IoU of approximately 27.3%). This reflects the inherent challenges of accurately segmenting morphologically complex and boundary-ambiguous lung cavities under the weakly supervised setting using only image-level labels and sparse scribbles. The current performance bottlenecks may stem from several aspects: firstly, the quality of the initial pseudo-labels is highly dependent on the CAM's ability to localize discriminative regions, which may be incomplete when the target exhibits high internal heterogeneity; secondly, the weak supervision signals themselves lack exhaustive boundary information, imposing a theoretical upper limit on the model's ability to learn fine contours; finally, the substantial variations in lesion size, morphology, and contrast across different datasets also pose difficulties for model generalization.

Regarding the prospects for clinical translation, we consider the current work primarily as a methodological validation. Although the results are encouraging, a series of more in-depth studies and validations are required before it could be directly deployed in clinical decision support systems. Examples include external validation on larger-scale, multi-center,

prospectively collected datasets; further quantitative analysis of the model's performance across different subgroups (e.g., based on cavity size, location, or etiology); and conducting necessary clinical trials to evaluate its practical impact on clinical workflows and decision-making. Furthermore, the model's computational efficiency is also a consideration for practical deployment, and future work could explore model lightweighting and inference acceleration techniques.

Overall, this study indicates that graph-based affinity learning holds considerable potential for enhancing segmentation performance under weak supervision and offers a promising research direction for reducing annotation costs in 3D medical image analysis. Future work will focus on improving absolute segmentation accuracy, enhancing cross-dataset generalization capability, and further evaluating its clinical relevance through larger-scale validation.

## Conclusion

In this paper, we have proposed a novel weakly supervised method for 3D semantic segmentation of lung cavities, achieving state-of-the-art performance across multiple benchmark datasets. Our method, which integrates graph-based affinity learning, region-wise affinity propagation, and scribble-based semantic segmentation, effectively addresses the challenges of weak supervision in medical image segmentation. The comprehensive ablation study demonstrated the critical role of each component in improving the segmentation accuracy, particularly in complex anatomical structures.While there are some limitations, such as high computational costs, our method shows potential as a reliable tool for the automated analysis of 3D medical images. Future improvements in annotation strategies and computational efficiency will further enhance its applicability and robustness in clinical settings.

## Author contributions

**Conceptualization:** Zhuoyi Tan, Zhengdong Li.

**Data curation:** Hizmawati Madzin.

**Formal analysis:** Zhuoyi Tan, Hizmawati Madzin, Zhengdong Li.

**Funding acquisition:** Zeyu Ding.

**Methodology:** Zeyu Ding, Zhuoyi Tan, Hizmawati Madzin, Zhengdong Li.

**Project administration:** Zhengdong Li.

**Software:** Zhuoyi Tan.

**Supervision:** Zeyu Ding, Hizmawati Madzin.

**Visualization:** Zhuoyi Tan, Juntao Liu.

**Writing – original draft:** Zhuoyi Tan, Hizmawati Madzin, Zhengdong Li.

**Writing – review & editing:** Zhuoyi Tan.

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
