## [Decision Letter · Decision Letter 0]

12 Dec 2025

PONE-D-25-44538

Refining Weak Supervision for Robust Lung Cavity Segmentation: A Graph-Affinity Framework with Boundary Constraints

PLOS One

Dear Dr. Tan,

Thank you for submitting your manuscript to PLOS ONE. After careful consideration, we feel that it has merit but does not fully meet PLOS ONE’s publication criteria as it currently stands. Therefore, we invite you to submit a revised version of the manuscript that addresses the points raised during the review process.

We look forward to receiving your revised manuscript.

Kind regards,

Hongchuan Yu

Academic Editor

PLOS One

**Journal Requirements:**

Please ensure that your manuscript meets PLOS ONE's style requirements, including those for file naming. The PLOS ONE style templates can be found at https://journals.plos.org/plosone/s/file?id=wjVg/PLOSOne_formatting_sample_main_body.pdf and https://journals.plos.org/plosone/s/file?id=ba62/PLOSOne_formatting_sample_title_authors_affiliations.pdf

**Additional Editor Comments:**

I invite 2 reviewers and their recommend: Both minor.

Reviewers have some concerns such as 1) novelty is weak, 2) dataset transparency, experimental reproducibility, and fair benchmarking...

Hope their comments are helpful for you to improve this work.

Finally, I will recommend "Minor" to this paper.

Reviewers' comments:

Reviewer's Responses to Questions

**Comments to the Author**

1. Is the manuscript technically sound, and do the data support the conclusions?

Reviewer #1: Yes

Reviewer #2: Yes

2. Has the statistical analysis been performed appropriately and rigorously?

Reviewer #1: Yes

Reviewer #2: No

3. Have the authors made all data underlying the findings in their manuscript fully available?

Reviewer #1: Yes

Reviewer #2: Yes

4. Is the manuscript presented in an intelligible fashion and written in standard English?

Reviewer #1: Yes

Reviewer #2: Yes

5. Review Comments to the Author

**Reviewer #1:** The manuscript addresses a clinically relevant and technically challenging task—weakly supervised lung cavity segmentation in CT imaging—by proposing a graph-based affinity network (GA-Net) with additional refinements such as EMA ensembling and a scribble-based boundary module. The topic aligns well with PLOS ONE’s scope in medical AI and image analysis. The writing is generally clear, and the technical presentation is well-structured. However, several critical issues related to clarity of contribution, dataset transparency, experimental reproducibility, and fair benchmarking must be resolved before the paper can be considered for publication.

Comments:

1. Clarity of Contribution

The paper repeatedly refers to the proposed work as a framework, but from the methodology and experiment sections, it is essentially a new method or model (GA-Net) rather than a broader framework.

Recommendation: Re-define the contribution precisely.

The abstract and introduction should explicitly highlight what is new compared to previous WSSS methods. Currently, the novelty statement is diffuse.

2. Literature Review and Dataset Exploration

The related work section (implied within the introduction) briefly mentions previous WSSS and medical segmentation works but does not adequately analyse key prior methods or datasets.

3. Dataset and Experimental Specification

The dataset description is too brief and does not indicate:

Imaging modalities (CT resolution, slice count),

Preprocessing steps (normalisation, lung mask extraction, augmentation),

How image-level or scribble labels were generated,

Hardware/software environment.

Some “processing seems missing,” e.g., how the pseudo-labels are derived and refined, and how scribbles are converted into supervision masks.

4. It is good to have a detailed subsection “Dataset and Preprocessing” and another “Implementation Details” under Methodology or Experiments.

5. Benchmarking and Fairness of Comparison

The results claim superiority over state-of-the-art WSSS techniques but provide no clear justification that comparisons are fair. It is good to include a concise Benchmark Configuration Table listing each compared method, supervision type, and training protocol to demonstrate fairness and consistency.

6. The paper does not analyse the computational cost or complexity of GA-Net. It is good to add a complexity discussion. It would strengthen the argument for practical robustness, particularly for clinical deployment.

7. Equations (1)–(9) are mathematically dense but lack intuitive explanation or pseudocode showing how the modules are trained end-to-end.

**Reviewer #2:** Overall, the manuscript appears to present a coherent and potentially valuable contribution to weakly supervised 3D lung cavity segmentation, with a plausible methodological rationale, multi-dataset evaluation, and ablations that generally support the main claims of incremental performance improvement. However, the current version would benefit from important clarifications and modest strengthening.

1. The manuscript mentions both fixed splits and 5-fold CV. Please clarify whether CV was limited to the training set with a held-out test set, and how hyper-parameters were chosen to prevent test leakage.

2. If the authors report significance, they should specify the tests and justify their assumptions. Fold-wise paired tests or bootstrap CIs for DSC/IoU would be more appropriate than t-tests on correlated training losses.

3. Key hyper-parameters (affinity thresholds, graph design, propagation settings) may strongly affect results; a brief robustness/sensitivity analysis would show the gains are not dependent on a narrowly tuned setup.

4. The gains over baselines are encouraging, but absolute DSC/IoU remain modest; the discussion should focus on relative improvements and avoid implying clinical readiness without further validation.

5. Add qualitative and quantitative subgroup analyses by lesion size, location, morphology, and severity to clarify where the method performs well and where it fails.

6. The manuscript is well organised and mostly clear, with generally standard English. However, the experimental protocol and parts of the methods would benefit from language polishing to improve clarity and consistency in terminology, acronyms, and the evaluation design.

7. The statistics support descriptive improvement claims, but the inferential evidence is not yet robust, particularly given the t-test usage and unclear CV design. Clearer protocol details and fold- or patient-level testing would strengthen the conclusions.

6. PLOS authors have the option to publish the peer review history of their article (what does this mean?). If published, this will include your full peer review and any attached files.

Reviewer #1: No

Reviewer #2: **Yes:** Armin Sheibanifard

---

## [Author Response · Author response to Decision Letter 1]

9 Jan 2026

Response to Reviewer:

Reviewer 1: The manuscript addresses a clinically relevant and technically challenging task—weakly supervised lung cavity segmentation in CT imaging—by proposing a graph-based affinity network (GA-Net) with additional refinements such as EMA ensembling and a scribble-based boundary module. The topic aligns well with PLOS ONE’s scope in medical AI and image analysis. The writing is generally clear, and the technical presentation is well-structured. However, several critical issues related to clarity of contribution, dataset transparency, experimental reproducibility, and fair benchmarking must be resolved before the paper can be considered for publication. Comments:

1. Clarity of Contribution The paper repeatedly refers to the proposed work as a framework, but from the methodology and experiment sections, it is essentially a new method or model (GA-Net) rather than a broader framework. Recommendation: Re-define the contribution precisely. The abstract and introduction should explicitly highlight what is new compared to previous WSSS methods. Currently, the novelty statement is diffuse.

Answer

The issues you raised regarding the clarity of our contributions and the accuracy of terminology are highly important. Following your suggestions, we have carefully revised the entire manuscript—particularly the Abstract and Introduction—to define our contributions more precisely and to highlight the core innovations more explicitly. The main revisions are summarized as follows: Clarification of the terminology “framework” vs. “method”:

We fully agree with your comment. The original manuscript used the term “framework” inappropriately, which may have caused conceptual ambiguity. In the revised version, we have consistently replaced it with “method” or “model” to better reflect the nature of our work. Specifically, our contribution is a novel weakly supervised segmentation method, centered on the proposed GA-Net model together with a set of supporting optimization strategies. Making the contributions explicit and highlighting novelty: We have rewritten key parts of the Abstract and Introduction to directly and clearly distinguish our approach from existing WSSS methods. The major improvements include:

In the Abstract, we adopted a more structured, point-wise presentation to clearly summarize our innovations at three levels.

In the Introduction, we added a dedicated “Main Contributions” subsection to summarize the core innovations in a structured (1–2–3) format.

Throughout the problem analysis and method description in the Introduction, we strengthened the “compared to existing methods” discussion by explicitly pointing out the limitations of conventional affinity propagation based on low-level features and global propagation mechanisms, and by explaining how our GA-Net and region-wise propagation are specifically designed to address these issues.

2. Literature Review and Dataset Exploration. The related work section (implied within the introduction) briefly mentions previous WSSS and medical segmentation works but does not adequately analyse key prior methods or datasets.

Answer

We fully agree that an in-depth analysis of prior work is essential for clearly positioning the contributions of this study. Following your suggestion, we have substantially expanded and rewritten the Introduction section (which also serves the role of reviewing related work) to provide a more systematic and critical discussion of key prior methods and the limitations of existing approaches and datasets. The main revisions and improvements are summarized as follows: Stronger and deeper analysis of representative WSSS methods:

Instead of only mentioning conventional approaches in a general manner, we now explicitly cite a series of representative methods (e.g., IRNet and related works) and provide a more detailed analysis of their core mechanisms (CAM-based seeds and affinity propagation) and inherent limitations. The revised text clearly explains that the traditional affinity propagation algorithms used in these methods (e.g., random walk) are fundamentally limited for medical lesion segmentation because the affinity matrices are typically constructed from local, low-level features. Such affinities are insufficient to model the long-range contextual relationships and structural dependencies required for irregular lung cavities, which often causes propagation failures around lesion boundaries and yields incomplete pseudo-labels. Clearer identification of limitations at the supervision-signal level: We have strengthened the discussion of weak supervision relying solely on image-level labels. In addition to noting that this paradigm encourages models to focus on the most discriminative local regions, we further analyze why subsequent efforts that introduce self-attention or boundary-aware constraints are still constrained by a critical bottleneck: pseudo-label quality in the early stages of training. This makes the motivation for introducing our scribble-guided supervision module more explicit and logically grounded. Tighter linkage between literature critique and task-specific challenges: All analyses are now explicitly anchored to the specific task of lung cavity segmentation. We clarify how the aforementioned general-purpose WSSS methods tend to fail under cavity-specific imaging characteristics—such as thin walls, internal septations, and highly irregular morphology—thereby making the related-work critique more targeted and relevant. As reflected in the revised manuscript, we have replaced the original relatively broad descriptions with more specific, better-referenced, and more critical content. We believe this revision significantly strengthens the depth of related-work discussion and provides a more solid foundation for clearly motivating the novelty of our approach.

3. Dataset and Experimental Specification

The dataset description is too brief and does not indicate: Imaging modalities (CT resolution, slice count), Preprocessing steps (normalisation, lung mask extraction, augmentation),

How image-level or scribble labels were generated, Hardware/software environment. Some “processing seems missing,” e.g., how the pseudo-labels are derived and refined, and how scribbles are converted into supervision masks.

Answer

Thank you for highlighting the importance of providing a more complete dataset description and experimental details. We fully agree that these elements are critical to ensure reproducibility and the reliability of the conclusions. Following each of your specific suggestions, we have performed a comprehensive and careful revision of the manuscript to substantially enrich the missing information. The main additions and revisions are summarized below:

Imaging modality and data preprocessing (new dedicated content):

In the “Data Preprocessing and Experimental Settings” section, we added a new paragraph titled “Imaging Modality and Parameters.” This paragraph explicitly describes the data sources and the ranges of original CT acquisition parameters (in-plane resolution, slice thickness, and number of slices). We also detail the unified preprocessing pipeline, including the resampling strategy, the CT intensity (HU) windowing range ([-1200, 600]), and the final normalization procedure, ensuring consistency and comparability across datasets.

Complete label generation and refinement pipeline (new dedicated subsection):

We added a new subsection titled “Label Generation and Refinement,” which systematically describes the full pipeline from the raw weak supervision signals to the final supervision masks used for training. This directly addresses your concern that some steps appeared to be missing. Specifically, we clarify:

Image-level annotation generation: We describe in detail how we constructed reliable image-level labels for public datasets lacking expert pixel-level annotations (e.g., ImageCLEF TB and TB Portals), including annotator qualifications, the annotation tool (ITK-SNAP), and the annotation protocol.

Pseudo-label derivation and refinement: We clearly present the full procedure starting from CAM generation, applying a dual-threshold strategy to identify confident regions, and then iteratively refining pseudo-labels using our proposed structure-aware affinity learning and region-wise affinity propagation.

Scribble generation and supervision: We explicitly clarify that our scribbles are automatically generated rather than manually drawn, derived from the contours of the refined pseudo-labels. These scribbles are then used to provide boundary-level supervision via a partial cross-entropy loss, addressing your question of how scribbles are converted into supervision masks.

Data augmentation strategy:

In the “Data Preprocessing and Experimental Settings” section, we added a paragraph on “Data Augmentation,” describing the specific online 3D augmentation techniques used (e.g., random 3D cropping and Gaussian noise perturbation). We also emphasize the principle of synchronizing spatial and intensity transformations to avoid introducing label noise.

Hardware and software environment:

At the end of the “Data Preprocessing and Experimental Settings” section, we added a paragraph titled “Hardware and Software Environment,” specifying key training configurations such as GPU model, memory, batch size, optimizer (AdamW), and learning rate scheduling (warm-up + cosine annealing).

Summary: Through these additions and revisions, we have systematically filled in all previously missing details related to dataset description, preprocessing, the label generation/refinement chain, and experimental configurations. We believe these improvements significantly enhance the rigor, transparency, and reproducibility of both the methods and experiments, providing sufficient information for readers and reviewers to evaluate the validity of our study.

4. It is good to have a detailed subsection “Dataset and Preprocessing” and another “Implementation Details” under Methodology or Experiments.

Answer:

We fully agree with your point that organizing “Datasets and Preprocessing” and “Implementation Details” as independent subsections can significantly improve the structure and clarity of both the Methods and Experiments, making it easier for readers (and future researchers) to reproduce our work. Following your suggestion, we have reorganized the Experiments section (according to the paper’s specific chapter arrangement) and created the following three well-structured subsections:

Datasets

This subsection provides a centralized and detailed description of the three publicly available tuberculosis chest CT datasets used in this study, including:

Data sources and task types: We explicitly state the origin of each dataset, the number of samples included, and its original task setting (classification or segmentation).

Data split strategy: We clearly report the exact number of images used for training and validation/testing in each dataset, along with the corresponding split ratios.

Key supervision information: By referring to Table~1, we summarize the cavity-related categorization criteria (e.g., cavity number and size) for each dataset, providing necessary context for the weak supervision setup.

Data Preprocessing and Experimental Setup

This subsection consolidates and expands the dataset-related descriptions in the original manuscript. It now systematically covers:

Dataset statistics and annotation types: An overview of the sample size and annotation type for each dataset.

Imaging modality and acquisition parameters: A clear description of CT acquisition parameters, resolution ranges, and the unified resampling and normalization pipeline.

Preprocessing pipeline: A complete and detailed preprocessing workflow, including HU clipping/windowing, normalization, and online data augmentation (e.g., 3D random cropping and Gaussian noise perturbation).

Weak-label generation pipeline: A concise summary of the full chain from raw image-level labels to pseudo-labels and finally to the scribble masks used for supervision (with detailed methodology described in the Methods section).

Implementation Details

This subsection is dedicated to all technical details related to model training and evaluation, including:

Hardware and software environment: Specific GPU model, memory, deep learning framework, etc.

Training configuration: Optimizer (AdamW), learning-rate schedule (warm-up + cosine annealing), batch size, number of epochs, and other hyperparameters.

Model architecture details: Key parameter settings for relevant modules (if applicable).

Evaluation metrics: The exact metrics used for comparison (e.g., Dice score and IoU) and how they are computed.

5. Benchmarking and Fairness of Comparison. The results claim superiority over state-of-the-art WSSS techniques but provide no clear justification that comparisons are fair. It is good to include a concise Benchmark Configuration Table listing each compared method, supervision type, and training protocol to demonstrate fairness and consistency.

Answer:

We fully agree that ensuring fairness and consistency in comparative experiments is essential, as it forms the basis for drawing reliable conclusions. Following your suggestion, we have made key additions to the “Comparison to State-of-the-Art” subsection to explicitly demonstrate the fairness of our comparisons. The specific revisions are as follows:

Added a benchmark configuration comparison table:

We have added Table 6 in the Results section, entitled “Benchmark configuration and fairness comparison across weakly supervised semantic segmentation methods.” This table provides a structured summary of all competing state-of-the-art methods included in our comparisons, along with their original references, the type of supervision used (Supervision Type), and the backbone architecture adopted (Backbone).

Clarified unified training settings across all baselines:

In the text below the table caption and in the main manuscript, we explicitly state that all compared methods were re-trained under exactly the same experimental settings, including a unified input patch size (128×128×96), optimizer (AdamW), and identical data preprocessing and augmentation pipelines. This design minimizes performance discrepancies caused by implementation differences and ensures that the reported improvements are attributable to methodological advances rather than inconsistent training conditions.

6. The paper does not analyse the computational cost or complexity of GA-Net. It is good to add a complexity discussion. It would strengthen the argument for practical robustness, particularly for clinical deployment.

Answer:

We fully agree that for any new method intended for clinical deployment, analyzing its computational cost and complexity is critical, as it is a key step in evaluating practical feasibility and robustness. Following your suggestion, we have added an independent subsection entitled “Computational Complexity Analysis” in the Experiments section, providing a comprehensive evaluation of the efficiency of GA-Net from both theoretical and empirical perspectives. The main contents are summarized as follows:

Theoretical complexity analysis:

We derive the theoretical computational complexity of GA-Net. By representing the image as a superpixel graph with Nnodes and Eedges, and analyzing the attention-based edge inference and node aggregation steps, we show that the complexity is O(L⋅E⋅d), where Ldenotes the number of graph layers and dis the feature dimension. Since the constructed graph is sparse (the number of edges Egrows linearly with the number of nodes N), the complexity of GA-Net scales linearly with image resolution, which provides a strong foundation for scalability to high-resolution clinical imaging.

Empirical runtime and memory consumption:

We measured practical performance on a standard 3D CT patch (128×128×96) using an NVIDIA RTX 3090Ti GPU. The results show that a single forward pass of GA-Net takes only ~42 ms, with a peak GPU memory usage of approximately 1.2 GB. Compared to a standard 3D U-Net baseline with a similar parameter count, GA-Net is about 2.9× faster and reduces memory consumpt

---

## [Editor Report · Decision Letter 1]

12 Jan 2026

Refining Weak Supervision for Robust Lung Cavity Segmentation: A Graph-Affinity Method with Boundary Constraints

PONE-D-25-44538R1

Dear Dr. Madzin,

We’re pleased to inform you that your manuscript has been judged scientifically suitable for publication and will be formally accepted for publication once it meets all outstanding technical requirements.

Kind regards,

Hongchuan Yu

Academic Editor

PLOS One

Additional Editor Comments (optional):

no further comments
---

## [Editor Report · Acceptance letter]

PONE-D-25-44538R1

PLOS One

Dear Dr. Madzin,

I'm pleased to inform you that your manuscript has been deemed suitable for publication in PLOS One. Congratulations! Your manuscript is now being handed over to our production team.

Kind regards,

on behalf of

Dr. Hongchuan Yu

Academic Editor

PLOS One